# Global variation in force-of-infection trends for human T*aenia solium* taeniasis/cysticercosis

**Matthew A Dixon**[1,2,3]*, **Peter Winskill**[2], **Wendy E Harrison**[3], **Charles Whittaker**[1,2], **Veronika Schmidt**[4,5], **Astrid Carolina Flórez Sánchez**[6], **Zulma M Cucunuba**[1,2†], **Agnes U Edia-Asuke**[7], **Martin Walker**[8], **María-Gloria Basáñez**[1,2]

[1]Department of Infectious Disease Epidemiology and London Centre for Neglected Tropical Disease Research (LCNTDR), Faculty of Medicine, School of Public Health, Imperial College London, London, United Kingdom; [2]MRC Centre for Global Infectious Disease Analysis, Department of Infectious Disease Epidemiology Faculty of Medicine, School of Public Health, Imperial College London, London, United Kingdom; [3]SCI Foundation, Edinburgh House, London, United Kingdom; [4]Department of Neurology, Center for Global Health, Technical University Munich (TUM), Munich, Germany; [5]Centre for Global Health, Institute of Health and Society, University of Oslo, Oslo, Norway; [6]Grupo de Parasitología, Instituto Nacional de Salud, Bogotá, Colombia; [7]Ahmadu Bello University, Zaria, Nigeria; [8]Department of Pathobiology and Population Sciences and London Centre for Neglected Tropical Disease Research (LCNTDR), Royal Veterinary College, Hatfield, United Kingdom

*For correspondence:
m.dixon15@imperial.ac.uk

Present address: †Departamento de Epidemiología Clínica y Bioestadística, Facultad de Medicina, Pontificia Universidad Javeriana, Bogotá, Colombia

**Competing interest:** The authors declare that no competing interests exist.

**Abstract** Infection by *Taenia solium* poses a major burden across endemic countries. The World Health Organization (WHO) 2021–2030 Neglected Tropical Diseases roadmap has proposed that 30% of endemic countries achieve intensified *T. solium* control in hyperendemic areas by 2030. Understanding geographical variation in age-prevalence profiles and force-of-infection (FoI) estimates will inform intervention designs across settings. Human taeniasis (HTT) and human cysticercosis (HCC) age-prevalence data from 16 studies in Latin America, Africa, and Asia were extracted through a systematic review. Catalytic models, incorporating diagnostic performance uncertainty, were fitted to the data using Bayesian methods, to estimate rates of antibody (Ab)-seroconversion, infection acquisition and Ab-seroreversion or infection loss. HCC FoI and Ab-seroreversion rates were also estimated across 23 departments in Colombia from 28,100 individuals. Across settings, there was extensive variation in all-ages seroprevalence. Evidence for Ab-seroreversion or infection loss was found in most settings for both HTT and HCC and for HCC Ab-seroreversion in Colombia. The average duration until humans became Ab-seropositive/infected decreased as all-age (sero) prevalence increased. There was no clear relationship between the average duration humans remain Ab-seropositive and all-age seroprevalence. Marked geographical heterogeneity in *T. solium* transmission rates indicate the need for setting-specific intervention strategies to achieve the WHO goals.

## Editor's evaluation

Dixon and colleagues have collated published "age-prevalence" data from 16 studies (4 from South America, 8 from Africa, and 4 from Asia) to estimate the force of infection of taeniasis/human cysticercosis across diverse endemic settings. This study addresses a major knowledge gap, as little is currently known regarding the extent of current/recent Taenia solium transmission worldwide. The

main limitation of this interesting study originates from the very nature of the primary data analyzed. Authors examine how the prevalence of (genus- but not species-specific) antigens shed in the stools or in the plasma and (more or less stage- and species-specific) antibodies changes across age groups and fit simple catalytic models to these very heterogeneous datasets.

## Introduction

The zoonotic cestode *Taenia solium* poses a substantial public health and economic challenge. Globally, *T. solium* is ranked as the foodborne parasitic infection contributing the highest number of disability-adjusted life-years (DALYs), an estimated 2.78 million DALYs in 2010 (*Havelaar et al., 2015*). Control and elimination of *T. solium* requires a One Health approach (*Thomas et al., 2019*; *Braae et al., 2019*) due to its complex multi-host life-cycle, involving infection with larval-stage cysticerci in the intermediate pig host (porcine cysticercosis, or PCC), infective stages (eggs) in the environment, and taeniasis (infection with the adult worm) in the definitive human host (human taeniasis, or HTT). When humans ingest *T. solium* eggs, establishment of cysticerci (human cysticercosis, or HCC) in the central nervous system results in neurocysticercosis. Neurocysticercosis is responsible for the major health burden associated with *T. solium*, causing approximately one third of epilepsy/seizure disorders in endemic settings (*Gripper and Welburn, 2017*).

Current interventions target PCC through mass treatment of pigs with oxfendazole and their vaccination with TSOL18, and HTT through mass treatment of humans with praziquantel or niclosamide. These control strategies have proven efficacious in field studies (*de Coster et al., 2018*; *Dixon et al., 2021*), but to date no large-scale interventions have been rolled out as part of national Neglected Tropical Disease (NTD) programmes, and therefore their impact alone or in combination in different epidemiological settings remains poorly understood and/or modelled (*CystiTeam Group for Epidemiology and Modelling of Taenia solium Taeniasis/Cysticercosis, 2019*). The milestones proposed in the new World Health Organization (WHO) NTD 2021–2030 roadmap (*Havelaar et al., 2015*; *World Health Organization, 2020*) focus on achieving intensified control in hyperendemic areas of 17 (27%) endemic countries. However, endemicity levels for *T. solium* have not been defined in terms of infection indicators and there exist limited data on the geographical distribution of *T. solium* prevalence at national scales. For other NTDs, pre-intervention levels of endemicity (according to infection prevalence, intensity, incidence or morbi-mortality) determine the magnitude, duration and likely success of control efforts (*NTD Modelling Consortium Onchocerciasis Group, 2019*). Therefore, successful *T. solium* intervention strategies will require tailoring to local epidemiological and socio-economic conditions that determine the intensity of transmission and will likely be informed by age- and/or sex-dependent contact rates and mixing patterns in endemic settings (*Dixon et al., 2021*; *CystiTeam Group for Epidemiology and Modelling of Taenia solium Taeniasis/Cysticercosis, 2019*; *Welburn et al., 2015*).

Global patterns in transmission rates have recently been explored for PCC by assessing the force-of-infection (FoI; the per-capita rate of infection acquisition) according to proposed endemicity levels (overall (sero)prevalence) and geography (*Dixon et al., 2020*). Estimates of antibody (Ab) and antigen (Ag) seroconversion (and seroreversion rates) for HCC have also been estimated from a small number of longitudinal surveys in Burkina Faso (*Dermauw et al., 2018*), Peru (*Garcia et al., 2001*), Ecuador (*Coral-Almeida et al., 2014*), and Zambia (*Mwape et al., 2013*), demonstrating substantial geographic variation.

Age-prevalence profiles are not only useful for estimating FoI and (sero)reversion rates, but also to provide insight into processes driving epidemiological dynamics, such as immunological responses or heterogeneity in exposure which may explain abrupt changes in HCC seropositivity in older individuals or distinct prevalence peaks in younger ages for HTT prevalence, as observed in the Democratic Republic of the Congo (DRC) (*Kanobana et al., 2011*; *Madinga et al., 2017*). Specific age-prevalence profiles may also provide insight into whether age-targeted interventions may be appropriate in certain settings. This study, therefore, and for the first time systematically collates and reviews all HTT and HCC age-(sero)prevalence data available from the literature and from collaborators, and uses simple and reversible catalytic models to estimate the FoI and Ab-seroreversion/infection loss rates from cross-sectional surveys. The results improve our understanding of global variation in the epidemiology of *T. solium* and provide estimates of transmission rates, crucial for guiding the design

of interventions. In addition, a country profile of FoI estimates is presented at the sub-national level for Colombia to exemplify how detailed FoI analyses can help understand local epidemiological patterns.

## Results

### Systematic review and study selection

After title, abstract and full-text eligibility screening of 236 studies initially identified (01/11/2014 to 02/10/2019), and 11 studies included in the *Coral-Almeida et al., 2015* literature review, a total of 16 studies were included in the analysis (PRISMA flowchart; *Figure 1*), originating from South America (n=4), Africa (n=8) and Asia (n=4) (*Figure 1—figure supplement 1*) and split by n=4 HTT and n=15 HCC surveys (full details in *Supplementary file 1*).

Total sample sizes were 6,653 individuals (range 576–4599; with individual age range of <1–96 years) across HTT surveys, which included three copro-Ag-based surveys and one Ab-based survey; 34,124 (125–29,360; cross-study age range of <1–95 years) across HCC-Ab surveys, and 12,934 (708–4993; cross-study age range of <1–96 years) across HCC-Ag surveys. Observed (sero)prevalence ranged from 4.5% to 23.4% (95% confidence interval (CI) range: 3.0–24.6%) for HTT surveys, 0.5–38.7% (0.1–41.6%) across HCC-Ab surveys, and 0.7–21.7% (0.5–24.5%) across HCC-Ag surveys.

Catalytic models (*Figure 2*; full details in Materials and methods) were fitted to the thus extracted (sero)prevalence data using a Bayesian framework that integrates uncertainty in diagnostic sensitivity and specificity from prior (published) information (*Supplementary file 1*). The FoI parameters of acquisition ($\lambda$) and reversion/loss ($\rho$) (for antibody datasets, a marker of exposure: $\lambda_{sero}$ and $\rho_{sero}$; for antigen datasets, a marker of active infection: $\lambda_{inf}$ and $\rho_{inf}$) were estimated for each dataset. The deviance information criterion (DIC) (*Spiegelhalter et al., 2002*) was used to compare individually- and jointly-fitted catalytic models.

We defined hyperendemic transmission settings as those with all-age *observed* HTT (sero)prevalence of ≥3% and all-age *observed* HCC (sero)prevalence of ≥6%. Studies with (sero)prevalence values below these were defined as endemic transmission settings. These putative endemicity definitions were defined following a literature review of studies referring to 'hyper' or 'highly' endemic settings (*Supplementary file 2* with additional explanatory text in Appendix 1).

### Global human taeniasis (HTT) copro-antigen and antibody seroprevalence

*Table 1* compares models fitted either including (reversible model) or excluding (simple model) HTT infection loss (when fitted to copro-Ag ELISA using the *Allan et al., 1990* protocol datasets, except in *Gomes et al., 2002* where a protocol is not specified) or Ab-seroreversion. For the copro-Ag ELISA datasets (to which models were jointly fitted to yield a single sensitivity and specificity posterior), DIC scores were similar (within one unit) between models with and without infection loss (*Table 1* and *Figure 3a*), indicating limited information to differentiate between model fits.

The copro-Ag ELISA datasets (*Gomes et al., 2002*; *Mwape et al., 2012*) were found in hyperendemic settings (all-age HTT (sero)prevalence ≥3%) in Zambia (6.3%) (*Mwape et al., 2012*) and Brazil (4.5%) (*Gomes et al., 2002*). For the models independently fitted to the single dataset in Lao People's Democratic Republic (Lao PDR) (*Holt et al., 2016*) using the rES33-immunoblot (*Wilkins et al., 1999*) and found in an endemic setting (all-age HTT antibody seroprevalence of 2.5%), there was also limited information to differentiate between model fits, with DIC scores similar (within one unit) between the model with and without Ab-seroconversion (*Table 1* and *Figure 3b*). The FoI ($\lambda_{sero}$) for the best-fit model to the rES33-immunoblot antibody dataset in Lao PDR (*Holt et al., 2016*), suggested a very low HTT Ab-seroconversion rate of 0.00046 year$^{-1}$ (all model fits and DIC scores in *Supplementary file 3*). The *Madinga et al., 2017* dataset in the DRC was omitted from the models jointly fitted across copro-Ag ELISA datasets, due to difficulty fitting catalytic models to such a distinct age-prevalence profile with a marked peak in early ages (see Discussion).

### Global human cysticercosis (HCC) antibody seroprevalence

HCC Ab-seroconversion with Ab-seroreversion (reversible model) provided an improved joint fit to the multiple datasets based on the antibody lentil lectin-purified glycoprotein enzyme-linked immunoelectrotransfer blot (LLGP-EITB) assay (*Tsang et al., 1989*; *Table 2* and *Figure 4a*). These fits were

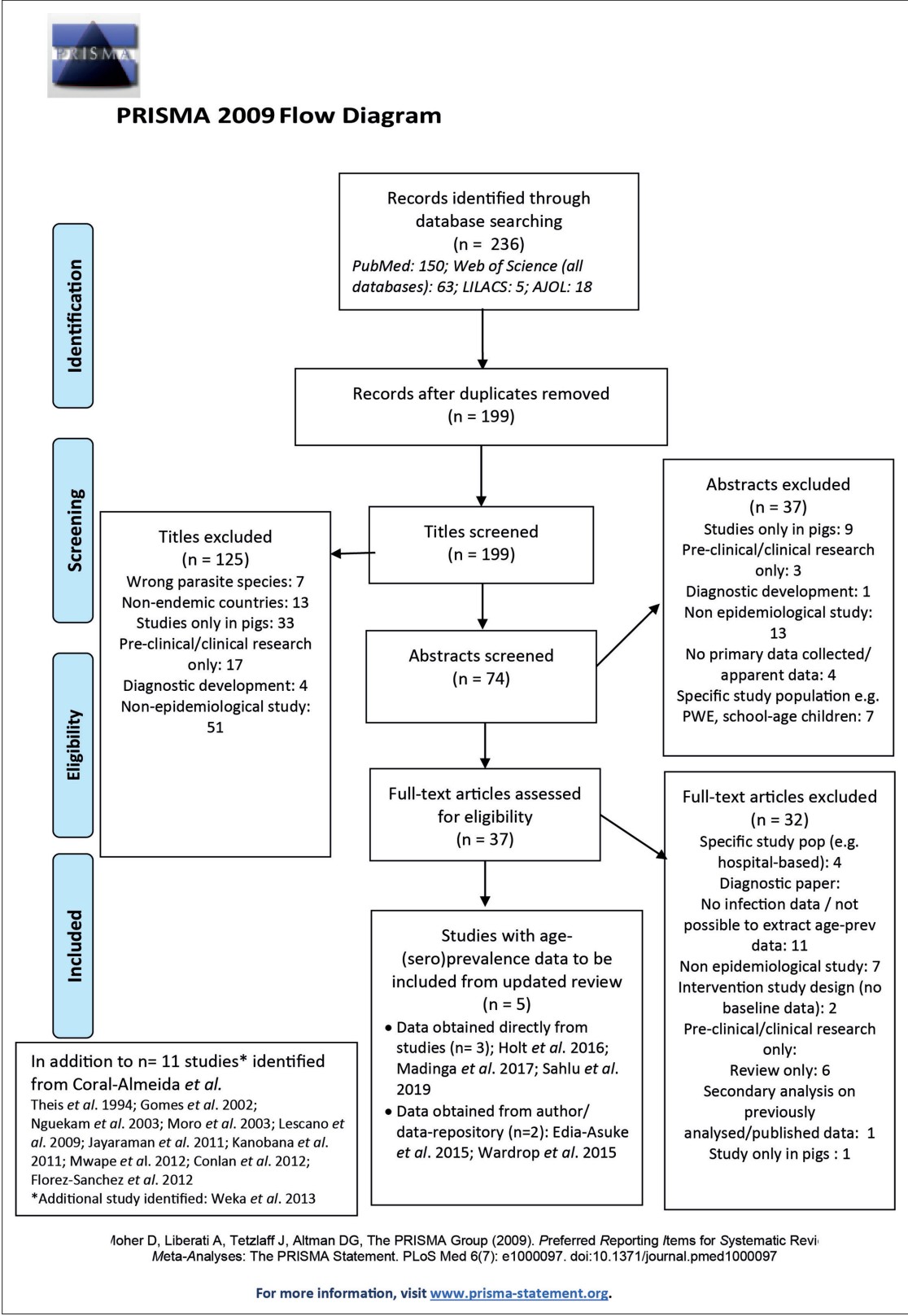

**Figure 1.** Published articles on age-(sero)prevalence profiles identified in the updated literature search using a **Moher et al., 2015** search. Additional studies identified from the **Coral-Almeida et al., 2015** review also shown. LILAC: Latin American and Caribbean Health Sciences Literature; AJOL: African Journals Online.

*Figure 1 continued on next page*

*Figure 1 continued*

The online version of this article includes the following figure supplement(s) for figure 1:

**Figure supplement 1.** Geographical distribution of studies with human taeniasis (HTT) and human cysticercosis (HCC) age-(sero) prevalence data included in the final analysis (n=16) by diagnostic target.

found in the proposed hyperendemic settings (≥6% HCC seroprevalence; with all-age seroprevalence from 12.7% to 24.7%) in Peru (*Lescano et al., 2009*; *Moro et al., 2003*), India (*Jayaraman et al., 2011*), and Bali (*Theis et al., 1994*), and an endemic setting in Brazil (1.6%) (*Gomes et al., 2002*). For the models jointly fitted to multiple datasets based on the IgG Ab-ELISA (*DiagAutom, 2016*) hyperendemic setting: all-age seroprevalence from 9.6% to 14.5% in Nigeria (*Edia-Asuke et al., 2015*; *Weka et al., 2013*), and the models independently fitted to the single dataset from Lao PDR (endemic setting: 3.0% all-age seroprevalence) (*Holt et al., 2016*) based on the rT24H-immunoblot (*Hancock et al., 2006*), the (simple) model without Ab-seroreversion provided an improved fit (*Table 2* and *Figure 4b and c*). *Supplementary file 4* presents all model fits and DIC scores.

## Global human cysticercosis (HCC) antigen seroprevalence

HCC infection acquisition with infection loss (reversible model) provided an improved fit for models fitted jointly to multiple datasets based on B158/B60 Ag-ELISA (*Brandt et al., 1992*; *Dorny et al., 2000*; *Table 3* and *Figure 5a*), found in one hyperendemic setting (all-age HCC seroprevalence of 21.7%) in the DRC (*Kanobana et al., 2011*), and endemic settings in Zambia (*Mwape et al., 2012*), Burkina Faso (*Sahlu et al., 2019*), Lao PDR (*Conlan et al., 2012*) and Cameroon (*Nguekam et al., 2003*; all-age HCC seroprevalences from 0.7% to 5.8%). For models fitted to the single dataset from a hyperendemic setting in Kenya (6.61%) (*Wardrop et al., 2015*), using the HP10 Ag-ELISA (*Harrison et al., 1989*), the (simple) model without infection loss provided an improved fit (*Table 3* and *Figure 5b*). *Supplementary file 5* presents all model fits and DIC scores.

## Country-wide analysis of human cysticercosis antibody seroprevalence trends in Colombia

HCC Ab-seroconversion with Ab-seroreversion (reversible model), fitted to multiple datasets using an IgG Ab-ELISA (*López et al., 1988*), provided an improved fit across the 23 (out of a total of 24) departments of Colombia (*Flórez Sánchez et al., 2013*) included in the analysis. *Supplementary file 6* presents parameter estimates from the best-fit reversible model by department (parameter estimates for the simple model fits by department can also be found in *Supplementary file 7*). *Figure 6* presents the best-fit Ab-seroconversion with Ab-seroreversion (reversible) model fit to each age-seroprevalence dataset across departments (*Figure 6—figure supplement 1* zooms into model fits in medium to lower all-age seroprevalence departments for improved resolution). One department (Bolívar) was omitted due to difficultly fitting to such a distinct age-seroprevalence profile (prevalence peak in early ages).

Figure 7 highlights the geographical variation in a HCC Ab-seroconversion or FoI, (**b**) HCC Ab-seroreversion rate and c the HCC antibody all-age seroprevalence, across the 23 departments. In addition, *Figure 7* presents substantial geographical variation in risk factors, including: (d) the proportion of individuals owning pigs, and (e) the proportion of individuals reporting open defecation practices by department (*Figure 7—figure supplement 1* highlights variation in the proportion of pigs being kept under free-ranging management practices and the proportion of pigs being kept under free-ranging/mixed practices in those owning pigs (n=3157)).

## Force-of-infection across settings

A more intuitive approach to understanding the FoI ($\lambda$) is to consider its reciprocal, which here corresponds to the average time until humans become Ab-seropositive ($1/\lambda_{sero}$) or infected ($1/\lambda_{inf}$) as inferred through Ag-seropositivity. Equally, the reciprocal of $\rho$ relates to the average duration that humans remain Ab-seropositive ($1/\rho_{sero}$) or infected ($1/\rho_{inf}$). Given the large number of estimates obtained for HCC, parameter estimates from best-fit models were compared across settings (by all-age (sero) prevalence of each dataset and by country). *Figure 8* shows an overall decline in the average time (in years) until humans become HCC Ab-seropositive or infected with increasing all-age HCC (sero)

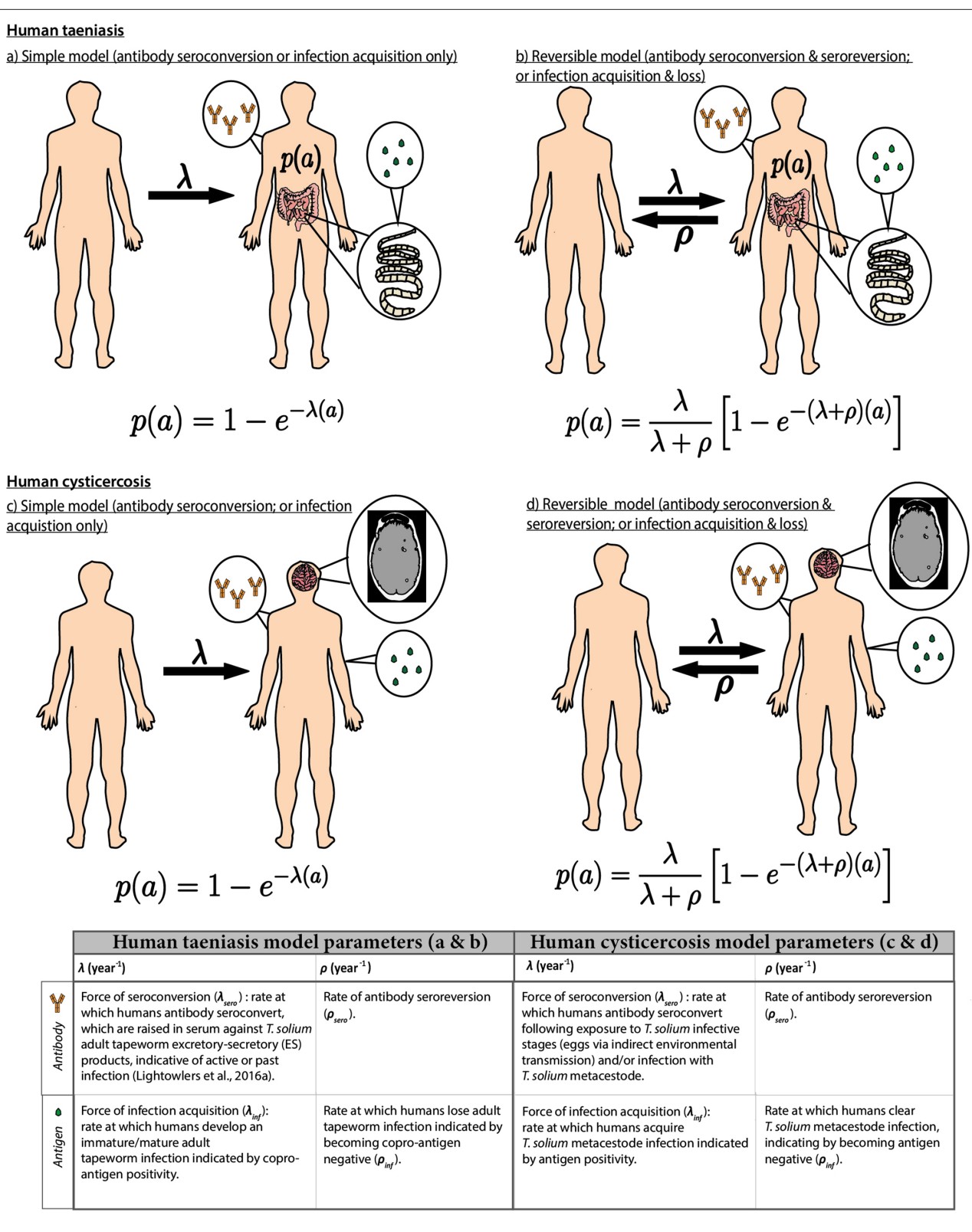

**Figure 2.** Simple and reversible catalytic model structure and equations fitted to data on the age (a)-specific (sero)prevalence (p(a)) data, where λ is the force-of-infection (rate of antibody (Ab)-seroconversion or infection acquisition) and ρ the rate of Ab-seroreversion or infection loss. The general mathematical form of the catalytic models (**equations 1a** to 1d) fitted to the human taeniasis (HTT)/human cysticercosis (HCC) Ab, HCC Ag and HTT copro-Ag prevalence datasets to estimate the prevalence (p) at human age (a). The saturating (sero)prevalence is given by $\lambda /(\lambda + \rho)$ which for the

*Figure 2 continued on next page*

*Figure 2 continued*

simple model is 100%, if the humans lived sufficiently long. The accompanying tables provide information on the definitions of the catalytic model parameters depending on the diagnostic method. Presence of adult tapeworm excretory-secretory products indicative of active or past HTT infection, as outlined by *Lightowlers et al., 2016a*.

prevalence, noting 18 studies identified in endemic settings (0.48–5.71% all-age HCC (sero)prevalence), and 19 estimates in hyperendemic settings (6.33–38.68% all-age HCC seroprevalence). Within countries, there was significant variation in times until humans become HCC Ab-seropositive or infected (*Figure 8—figure supplement 1a*).

*Figure 8b* presents strong evidence for similar average durations of remaining Ab-seropositive ($1/\rho_{sero}$) across different endemicity settings and countries (*Figure 8—figure supplement 1b*). However, there was evidence for a trend of decreasing duration of humans remaining infected ($1/\rho_{inf}$) with increasing all-age prevalence (antigen-based studies; n=5). *Figure 8—figure supplement 2* shows the relationship between Ab-seroconversion and Ab-seroreversion rates, and the relationship between infection acquisition rates and infection loss rates.

## Discussion

This paper presents the first global FoI estimates for *T. solium* HTT and HCC, identifying geographical heterogeneity that supports calls to implement both locally-adapted control efforts (*Johansen et al., 2017Gabriël et al., 2017*), and setting-specific parameterisations of *T. solium* transmission models to support such efforts (*Dixon et al., 2019*). We have also placed our HTT FoI estimates in the context of (putative) endemic or hyperendemic settings, with 0.44 per 1,000 people per year for a (single) endemic setting, and from 9.6 to 21 per 1000 per year for two hyperendemic settings. HCC

**Table 1.** Parameter posterior estimates for the best-fit catalytic models fitted to human taeniasis age-(sero)prevalence datasets (ordered by decreasing all-age (sero)prevalence).

Parameters estimated from antibody-based datasets measure exposure dynamics, with seroconversion $\lambda_{sero}$ and seroreversion $\rho_{sero}$ rates. Parameters estimated from antigen-based datasets measure active infection dynamics, with infection acquisition $\lambda_{inf}$ and infection loss $\rho_{inf}$ rates.

| Dataset; Country | All-age observed (sero)- prevalence (%) (95% CI)* | Catalytic model | Diagnostic sensitivity (95% BCI) | Diagnostic specificity (95% BCI) | $\lambda$=infection acquisition ($\lambda_{inf}$) or Ab-seroconversion ($\lambda_{sero}$) rate, year$^{-1}$ (95% BCI)† | $\rho$=infection loss ($\rho_{inf}$) or Ab-seroreversion ($\rho_{sero}$) rate, year$^{-1}$ (95% BCI)† |
|---|---|---|---|---|---|---|
| Models jointly fitted to multiple datasets (Copro-Ag ELISA) ‡ | | | | | | |
| *Mwape et al., 2012*; Zambia Antigen § | 6.32 (4.65–8.37) | | | | 0.021 (0.0038–0.062) | 0.768 (0.362–0.991) |
| *Gomes et al., 2002*; Brazil Antigen** | 4.51 (2.97–6.54) | Reversible ¶ | 0.824 (0.533–0.972) | 0.959 (0.941–0.976) | 0.0096 (0.00072–0.032) | 0.731 (0.379–0.978) |
| Models independently fitted to single datasets (rES33-immunoblot)†† | | | | | | |
| *Holt et al., 2016*; Lao PDR Antibody | 2.49 (1.51–3.87) | Simple ‡‡ | 0.982 (0.959–0.996) | 0.992 (0.978–0.999) | 0.00044 (0.000103–0.00082) | NA |

*Observed (sero)prevalence data are accompanied by 95% confidence intervals (95% CI) calculated by the Clopper-Pearson exact method.

†Parameter median posterior estimates are presented with 95% Bayesian credible intervals (95% BCI).

‡Diagnostic sensitivity and specificity jointly fitted for the Copro-Ag ELISA (*Allan et al., 1990*) across age-prevalence datasets; model parameters interpreted in terms of representing infection, with infection acquisition ($\lambda_{inf}$) and infection loss ($\rho_{inf}$) rates.

§Based on the *Allan et al., 1990* protocol.

¶Best-fitting model determined by DIC (models jointly fitted to multiple dataset).

**no protocol specified (assumed to be the *Allan et al., 1990* protocol). NA = Not applicable;, PDR: People's Democratic Republic.

††Model parameters for antibody-based age-seroprevalence data (based on the rES33-immunoblot *Wilkins et al., 1999*), interpreted in terms of exposure, with Ab-seroconversion ($\lambda_{sero}$) and Ab-seroreversion ($\rho_{sero}$) rates.

‡‡Best-fitting model determined by DIC (models independently fitted to single dataset).

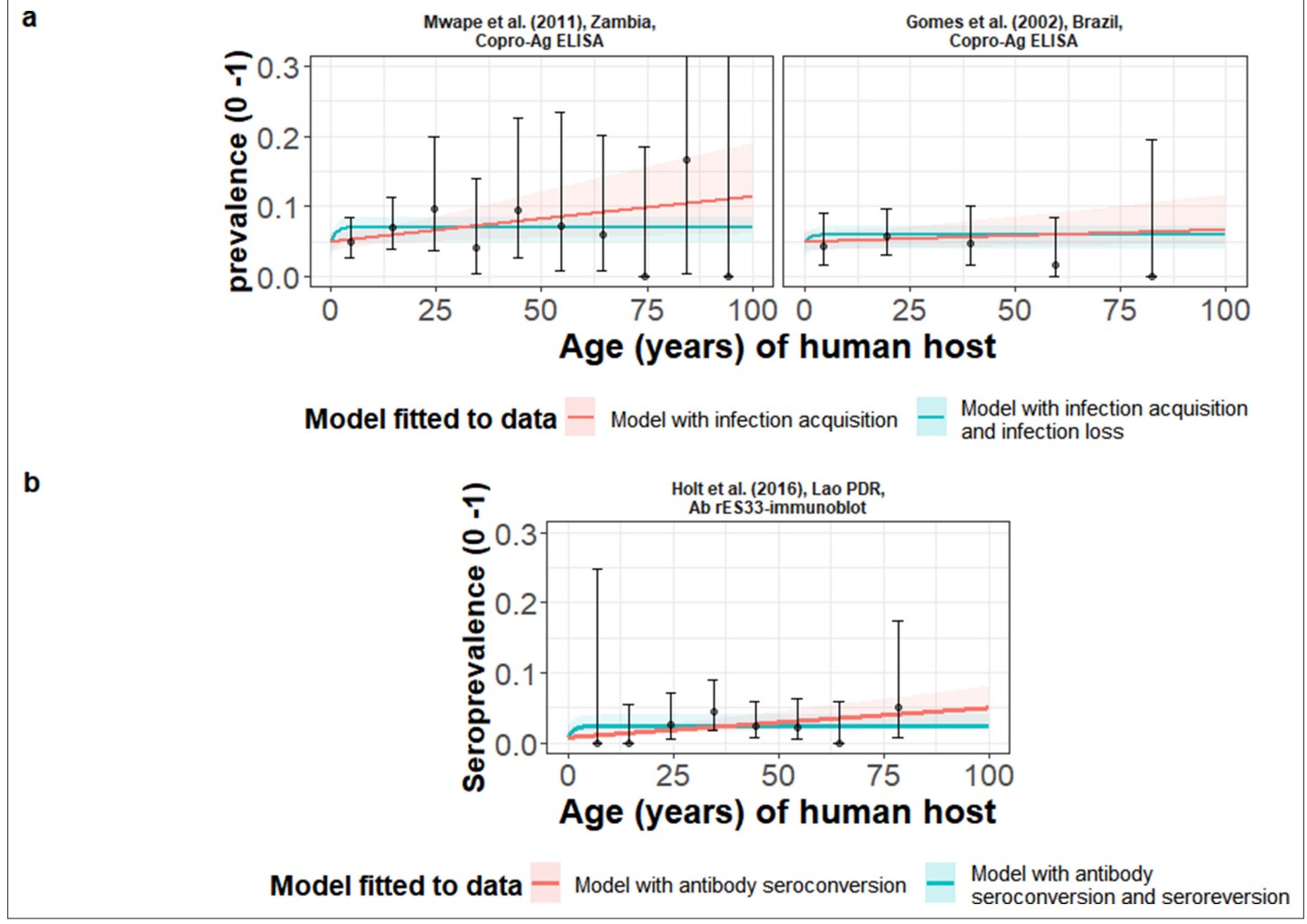

**Figure 3.** The relationship between human taeniasis copro-antigen (Copro-Ag) and antibody (sero)prevalence and human age (in years) for each dataset. Human taeniasis (HTT) infection acquisition (simple) or acquisition with infection loss (reversible) catalytic models jointly fitted to multiple datasets (where single diagnostic sensitivity and specificity values were estimated; dataset-specific $\lambda$ and $\rho$ estimates were obtained) in **a**; HTT antibody (Ab)-seroconversion (simple) or Ab-seroconversion with Ab-seroreversion to a single dataset in **b**. 95% confidence intervals associated with observed (sero)prevalence point estimates are also presented. Bayesian Markov chain Monte Carlo methods were used to fit the models to data, with the parameter posterior distributions used to construct predicted (all age) (sero)prevalence curves and associated 95% Bayesian credible intervals (BCIs). Best-fitting model selected by deviance information criterion (DIC); both models presented if difference between DIC < 2 (both models have similar support based on the data); a difference > 10 units indicates that the models are significantly different and therefore only superior fitting model (lowest DIC) is presented. The non-zero predicted (sero)prevalence at age 0 is due to less than 100% specificity for all tests. The 95% confidence intervals (95% CI) for age-(sero)prevalence data-points are calculated by the Clopper-Pearson exact method.

FoI estimates ranged from 0.086 to 21 per 1000 per year in (18, including Colombian departments) endemic settings (0.086–21 for Ab-based surveys; 0.17–4.4 for Ag-based surveys), and from 0.54 to 120 per 1000 per year in (19) hyperendemic settings (2.3–120 for Ab-based surveys; 0.54–110 for Ag-based surveys). Further work will be required to refine the proposed (and preliminary) characterisation of endemicity levels, perhaps linked to severity/morbidity as in other NTDs (*Prost et al., 1979*; *Smith et al., 2013*). This will be relevant to inform what constitutes a hyperendemic setting in need of intensified control interventions, as proposed in the WHO 2021–2030 goals for *T. solium* (*Havelaar et al., 2015*; *World Health Organization, 2020*). Our work also follows the five principles distilled by *Behrend et al., 2020* for communicating policy-relevant NTD modelling to stakeholders and implementation partners (*Supplementary file 8*).

Across epidemiological settings, there was a clear trend of decreasing average time until humans become HCC Ab-seropositive or infected as inferred through Ag-seropositivity (the reciprocal of

**Table 2.** Antibody seroprevalence and parameter estimates for the best-fit catalytic models fitted to each observed human cysticercosis (antibody) age-seroprevalence dataset (ordered by decreasing all-age seroprevalence). Antibody seroconversion and seroreversion rates represent markers of exposure.

| Dataset; Country | All-age observed sero- prevalence (%) (95% CI)* | Catalytic model | Diagnostic sensitivity (95% BCI) | Diagnostic specificity (95% BCI) | $\lambda_{sero}$sero seroconversion rate, year$^{-1}$ (95% BCI)† | $\rho_{sero}$sero seroreversion rate, year$^{-1}$ (95% BCI)† |
|---|---|---|---|---|---|---|
| **Models jointly fitted to multiple datasets (LLGP-EITB assay) ‡** | | | | | | |
| *Lescano et al., 2009*; Peru | 24.66 (21.59–27.94) | | | | 0.12 (0.067–0.19) | 0.38 (0.21–0.62) |
| *Moro et al., 2003*; Peru | 20.82 (16.48–25.71) | | | | 0.11 (0.0504–0.21) | 0.501 (0.23–0.92) |
| *Jayaraman et al., 2011*; India | 15.81 (13.66–18.16) | | | | 0.019 (0.0095–0.093) | 0.105 (0.042–0.56) |
| *Theis et al., 1994*; Bali | 12.68 (10.48–15.16) | | | | 0.024 (0.011–0.052) | 0.16 (0.054–0.38) |
| *Gomes et al., 2002*; Brazil | 1.64 (0.82–2.93) | Reversible§ | 0.976 (0.937–0.994) | 0.980 (0.967–0.988) | 0.000086 (0.000011–0.00066) | 0.43 (0.098–1.49) |
| **Models jointly fitted to multiple datasets (IgG Ab-ELISA) ¶** | | | | | | |
| *Edia-Asuke et al., 2015*; Nigeria | 14.53 (10.72–19.06) | | | | 0.0044 (0.0018–0.0064) | NA |
| *Weka et al., 2013*; Nigeria | 9.60 (5.06–16.17) | Simple§ | 0.872 (0.784–0.942) | 0.974 (0.916–0.998) | 0.0023 (0.00053–0.0046) | NA |
| **Models independently fitted to an single dataset (rT24H-immunoblot)** | | | | | | |
| *Holt et al., 2016*; Lao PDR | 2.96 (1.86–4.44) | Simple†† | 0.964 (0.914–0.988) | 0.986 (0.969–0.997) | 0.00044 (0.000049–0.00090) | NA |

*Observed seroprevalence data are accompanied by 95% confidence intervals (95% CI) calculated by the Clopper-Pearson exact method.

†Parameter median posterior estimates are presented with 95% Bayesian credible intervals (95% BCI). NA = Not applicable; PDR: People's Democratic Republic.

‡Diagnostic sensitivity and specificity jointly fitted for the LLGP-EITB assay (*Tsang et al., 1989*) across datasets.

§Best-fitting model determined by DIC (models jointly fitted to multiple datasets).

¶Diagnostic sensitivity and specificity jointly fitted for the IgG Ab-ELISA (DiagnosticAutomation/CortezDiagnostic, Inc, 2016) across datasets.

**Best-fitting model determined by DIC (models independently fitted to a single dataset).

††Based on the rT24H-immunoblot (*Hancock et al., 2006*).

$\lambda_{sero}$ or $\lambda_{inf}$) with increasing all-age seroprevalence. This makes intuitive sense as the FoI increases in settings with more intense transmission (reflected by a higher all-age seroprevalence). It was not possible to discern whether such trends exist for HTT given the limited availability of HTT datasets with age-stratified prevalence data. Infection loss with infection acquisition was indicated for two HTT copro-Ag-ELISA-based surveys, and five of six HCC antigen-based surveys, supporting inclusion of parameters representing recovery from HTT and HCC in *T. solium* transmission models (*Dixon et al., 2019*). Our infection loss rates for HTT indicate that the duration of adult tapeworm infection (the reciprocal of $\rho_{inf}$) ranges from 1.3 to 1.4 years (with uncertainty bounds from 1.0 to 2.8 years), aligning with literature estimates suggesting that adult tapeworms live for less than 5 years (*Garcia et al., 2014*), although our values are only based on two studies. In addition, reinfection of individuals with the adult tapeworm is also likely to occur, particularly in high-endemicity settings; therefore, the persistence of antibodies against the adult worm is likely to complicate the measurement of reinfection rates (where antibody seroconversion is equated to infection, although we take care to differentiate between these two processes when interpreting the $\lambda_{sero}$ and $\lambda_{inf}$ parameters). However, with the limited number of HTT-based surveys available to estimate antibody seroconversion and duration of antibody parameters, it is difficult to determine to what extent this is an issue. Understanding antibody kinetics in relation to spontaneous death and elimination of the adult tapeworm (either naturally or following treatment) is therefore an important area for further consideration, especially towards understanding whether seroreversion rates are related to baseline antibody levels. HCC Ab-seroreversion rates (or infection loss rates) from this analysis are also consistently larger than HCC Ab-seroconversion rates (or infection acquisition rates), in agreement with comparisons of HCC Ab-seroreversion

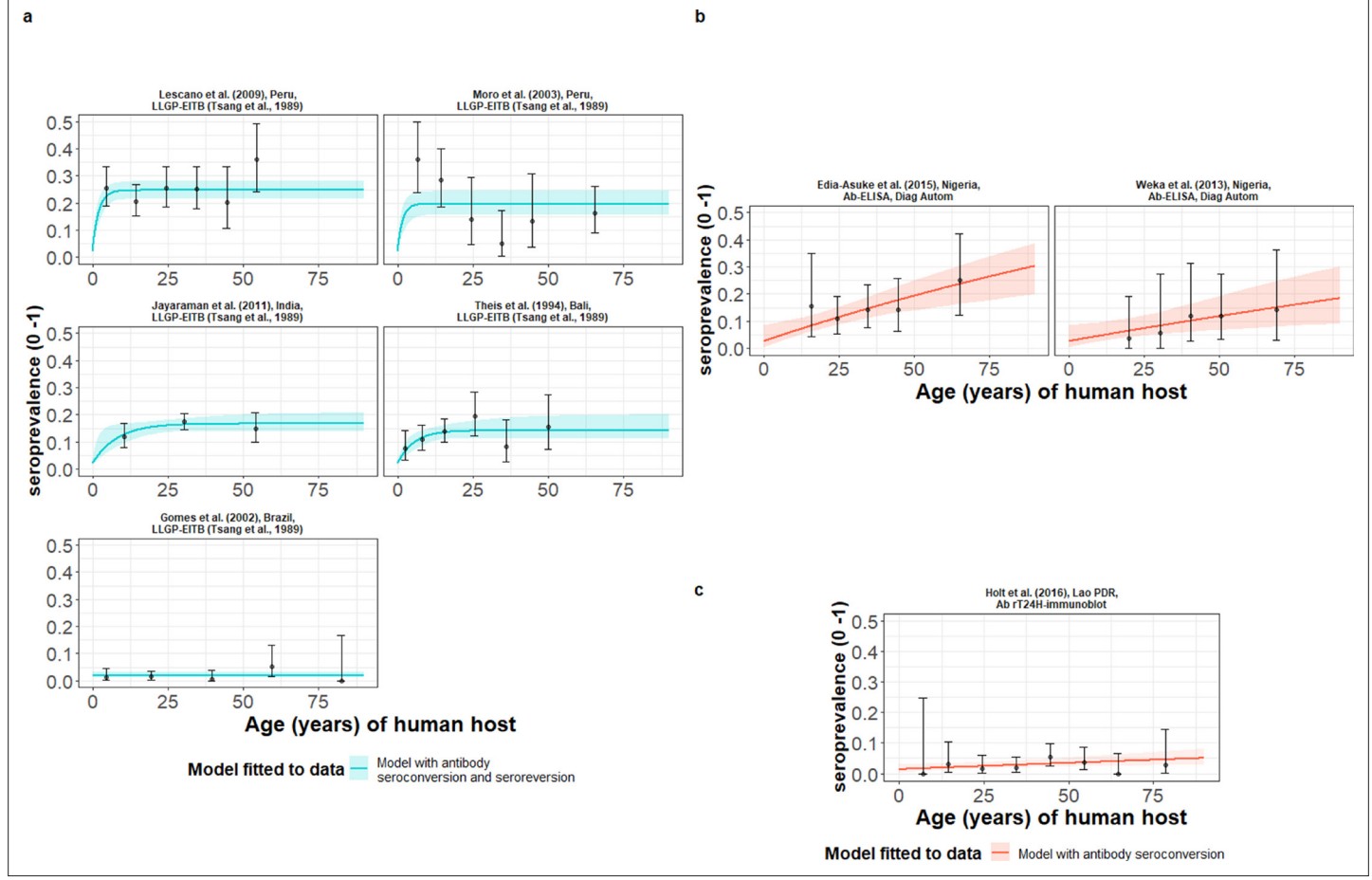

**Figure 4.** The relationship between human cysticercosis antibody (Ab)-seroprevalence and human age (in years) for each dataset. Ab-seroconversion (simple) or Ab-seroconversion with Ab-seroreversion (reversible) catalytic models (**a & b**) jointly fitted to multiple datasets (single diagnostic sensitivity and specificity values estimated; dataset-specific $\lambda_{sero}$ and $\rho_{sero}$ estimates obtained) and (**c**) models independently fitted to a single dataset, including 95% confidence intervals associated with observed Ab-seroprevalence point estimates. Bayesian Markov chain Monte Carlo methods were used to fit the models to data, with the estimated parameter posterior distributions used to construct predicted (all age) seroprevalence curves and associated 95% Bayesian credible intervals (BCIs). Best-fitting models were selected using the deviance information criterion (DIC); both models presented if difference between DIC < 2 (both models have similar support based on the data); a difference > 10 units indicates that the models are significantly different and therefore only superior fitting model (lowest DIC) is presented. The non-zero predicted seroprevalence at age 0 is due to less than 100% specificity for all tests. The 95% confidence intervals (95% CIs) for age-seroprevalence data-points are calculated by the Clopper-Pearson exact method.

to Ab-seroconversion (or Ag-seroreversion to Ag-seroconversion) estimates in the literature, which are generally based on cumulative seroconversion and seroreversion proportions, or rule-based modelling approaches (*Dermauw et al., 2018*; *Garcia et al., 2001*; *Coral-Almeida et al., 2015*; *Mwape et al., 2013*).

Some of the modelled estimates presented here indicate large differences between FoI and Ab-seroreversion/infection loss rates. These differences are particularly evident in datasets, such as Brazil (*Gomes et al., 2002*) and Risaralda Department in Colombia (*Flórez Sánchez et al., 2013*), where the reversible model was selected for flat age-(sero)prevalence profiles in low (sero)prevalence settings, where the seroreversion parameter provided minimal information with large uncertainty. The flat age-(sero)prevalence in these settings may also be likely explained by false-positives generated given the slightly sub-optimal performance of the serological antibody diagnostic used e.g. posterior median specificity estimates of 98% in Brazil (*Gomes et al., 2002*) for the antibody LLGP-EITB test (*Tsang et al., 1989*), highlighting issues with imperfect specificity of diagnostics in settings with very low transmission intensity.

The high cysticercosis Ab-seroreversion rates indicated across datasets are likely indicative of substantial transient responses generated from exposure without establishment of infection. This may

**Table 3.** Antigen seroprevalence and parameter estimates for the best-fit catalytic models fitted to each observed human cysticercosis (antigen) age-seroprevalence dataset (ordered by decreasing all-age seroprevalence).
Antigen-based infection acquisition and infection loss rates represent markers of active infection.

| Dataset; Country | All-age observed seroprevalence (%) (95% CI)* | Catalytic model | Diagnostic sensitivity (95% BCI) | Diagnostic specificity (95% BCI) | $\lambda_{inf}$ inf infection acquisition rate, year$^{-1}$ (95% BCI)† | $\rho_{inf}$ inf infection loss rate, year$^{-1}$ (95% BCI)† |
|---|---|---|---|---|---|---|
| Models jointly fitted to multiple datasets (the B158/B60 Ag-ELISA) ‡ | | | | | | |
| *Kanobana et al., 2011*; DRC | 21.66 (19.01–24.49) | | | | 0.11 (0.077–0.17) | 0.330 (0.246–0.464) |
| *Mwape et al., 2012*; Zambia | 5.79 (4.19–7.77) | | | | 0.0044 (0.0027–0.0088) | 0.023 (0.001–0.085) |
| *Sahlu et al., 2019*; Burkina Faso | 2.45 (1.93–3.08) | | | | 0.0016 (0.00091–0.0032) | 0.029 (0.0016–0.11) |
| *Conlan et al., 2012*; Lao PDR | 2.22 (1.49–3.17) | | | | 0.0018 (0.00073–0.0034) | 0.063 (0.0076–0.12) |
| *Nguekam et al., 2003*; Cameroon | 0.68 (0.47–0.95) | Reversible § | 0.909 (0.810–0.967) | 0.999 (0.995–0.999) | 0.00017 (0.000077–0.00030) | 0.004 (0.00018–0.026) |
| Models independently fitted to a single dataset (the HP10 Ag-ELISA) ¶ | | | | | | |
| *Wardrop et al., 2015*; Kenya | 6.61 (5.57–7.76) | Simple** | 0.850 (0.735–0.927) | 0.944 (0.930–0.959) | 0.00054 (0.000053–0.0013) | NA |

NA = Not applicable;.DRC: Democratic Republic of the Congo; PDR: People's Democratic Republic.

**Observed seroprevalence data are accompanied by 95% confidence intervals (95% CI) calculated by the Clopper-Pearson exact method.

†Parameter median posterior estimates are presented with 95% Bayesian credible intervals (95% BCI).

‡Diagnostic sensitivity and specificity jointly fitted for the B158/B60 Ag-ELISA (*Brandt et al., 1992*; *Dorny et al., 2000*).

§Best-fitting model determined by DIC (models jointly fitted to multiple datasets).

¶Based on the HP10 Ag-ELISA (*Harrison et al., 1989*).

**Best-fitting model determined by DIC (models independently fitted to a single dataset).

also explain the high all-age Ab-seroprevalence estimates observed in field studies (*Garcia et al., 2001*). Even with Ag-based surveys, a marker of infection rather than exposure, Ag-positivity may still indicate some degree of transient responses due to partial establishment of (immature) cysts or establishment followed by rapid degeneration of (mature) cysts (*Mwape et al., 2012*). One expectation was that Ab-seroreversion and infection loss rates are fundamentally biological processes so would be fairly consistent across settings. Since Ab-seroreversion or infection loss parameters were allowed to vary among settings, the consistency of the posterior estimates for these parameters indicates that rates of HCC Ab-seroreversion are stable (i.e. no relationship between all-age Ab-seroprevalence and the average duration humans remain Ab-seropositive, *Figure 8b*; no relationship between the Ab-seroconversion and Ab-seroreversion rates, *Figure 8—figure supplement 2*), confirming this a priori expectation. The average duration humans remain infected (i.e., the reciprocal of the HCC infection loss rate), however, decreases with increasing seroprevalence (and a positive relationship between infection acquisition rates and infection loss rates, *Figure 8—figure supplement 2*). Such a signal seen for Ag-based estimates, albeit with only 5 observations, may indicate increased transient responses in higher transmission settings due to more partial establishment of infection. There is however an absence of wider literature to support the hypothesis of exposure intensity affecting probability of successful parasite establishment, but such (transmission intensity-dependent) parasite establishment processed have been incorporated into transmission models for other NTDs (*Hamley et al., 2019*), highlighting an area of future research with implications for control efforts. In the case of Ab-seroreversion rates, where more data are available compared to infection loss rates, it may be suitable in future work to jointly fit the $\rho_{sero}$ parameter for reversible models across settings.

This analysis also provides, for the first time, a detailed country-level analysis, focused on Colombia, exploring the geographical variation of key epidemiological metrics (FoI and antibody seroreversion) alongside understanding potential drivers of these trends, using a rich dataset generated by *Flórez*

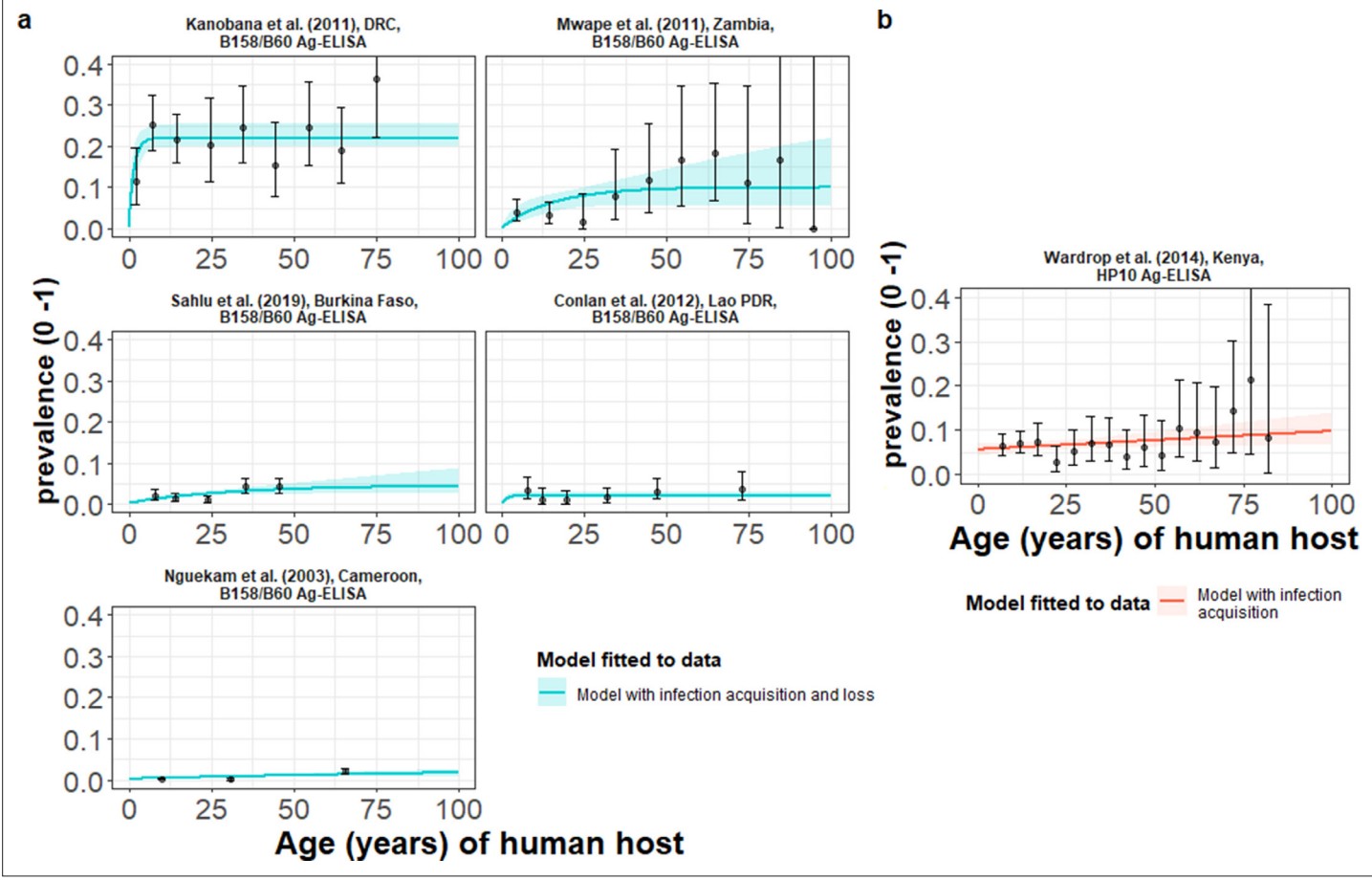

**Figure 5.** The relationship between human cysticercosis antigen (Ag)-seroprevalence and human age (in years) for each dataset. Infection acquisition (simple) or infection acquisition and loss (reversible) catalytic models (**a**) jointly fitted to multiple datasets (single diagnostic sensitivity and specificity values estimated; dataset-specific $\lambda_{inf}$ and $\rho_{inf}$ estimates obtained) and (**b**) models independently fitted to a single dataset, including 95% confidence intervals associated with observed Ag-seroprevalence point estimates. Bayesian Markov chain Monte Carlo methods were used to fit the models to data, with the parameter posterior distributions used to construct predicted (all age) seroprevalence curves and associated 95% Bayesian credible intervals (BCIs). Best-fitting model selected by deviance information criterion (DIC); both models presented if difference between DIC < 2 (both models have similar support based on the data); a difference > 10 units indicates that the models are significantly different and therefore only superior fitting model (lowest DIC) is presented. The non-zero predicted seroprevalence at age 0 is due to less than 100% specificity for all tests. The 95% confidence intervals (95% CI) for age-seroprevalence data-points are calculated by the Clopper-Pearson exact method.

*Sánchez et al., 2013*. Our results identify substantial subnational variation in HCC FoI (measured by Ab-seroconversion) and Ab-seroreversion rates. There appears to be only a limited relationship between these epidemiological quantities, Ab-seroprevalence and other risk factors, including open defecation and pig ownership at the department level, highlighting the complex nature of exposure drivers for HCC. For example, the southeastern rural departments of Vaupés and Amazonas show the highest all-age seroprevalence and FoI estimates. But while Vaupés has one of the highest reported proportions of open defecation, pig ownership is low in both departments. Notwithstanding, substantial subnational variation indicates the need for tailored subnational approaches to control. Spatial variation has been explored formally for this dataset by including spatial structure (i.e. spatial correlation in seropositivity estimated up to approximately 140 km at the Municipality level) into a risk factor analysis, and identifying hot spots of unexplained residual clustering in northern and southern municipalities in Colombia (*Galipó et al., 2021*). The findings from this study further supports the requirement for designing subnational and targeted interventions (risk groups such as those with low education levels and displaced persons), alongside the need to understand epidemiological dynamics at the subnational level.

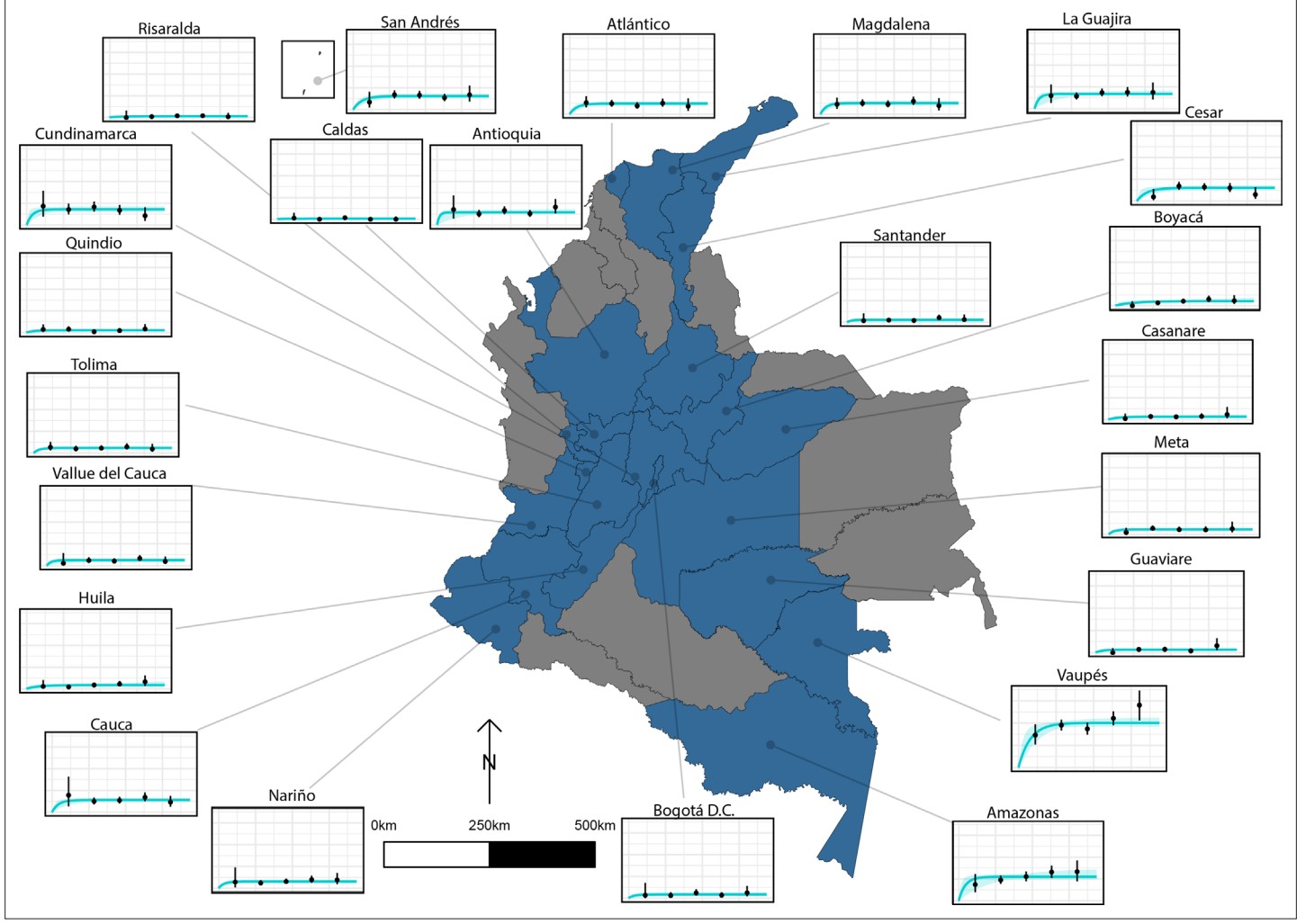

**Figure 6.** The relationship between human cysticercosis antibody (Ab)-seroprevalence and human age (in years) for 23 departments in Colombia. Each graph shows observed human cysticercosis Ab age-seroprevalence data (black points) and fitted reversible model (Ab-seroconversion with Ab-seroreversion; best-fitting model). Y-axis units are from 0 to 1 (HCC Ab-seroprevalence), with major y-axis gridlines at 0, 0.2, 0.4, and 0.6 seroprevalence. X-axis units are from 0 to 80 years (human age), with major x-axis gridlines at 0, 20, 40, 60, and 80 years of age.

The online version of this article includes the following figure supplement(s) for figure 6:

**Figure supplement 1.** Human cysticercosis antibody (Ab) age-seroprevalence profiles and (selected by Deviance Information Criterion) model with seroconversion and seroreversion model fits to (**a**) medium all-age Ab-seroprevalence Colombian Departments (6.33–14.37%) and (**b**) low all-age Ab-seroprevalence Colombian Departments (0.48–4.92%).

**Figure supplement 2.** Human cysticercosis antibody serosurvey sample size by department in Colombia.

**Figure supplement 3.** Colombia (**a**) age- antibody (Ab) seroprevalence profile stratified by sex for each department (to determine whether profiles clearly differ by sex) and compared to (**b**) age- Ab-seroprevalence (non-sex stratified) profiles for all 24 departments.

A potential limitation of this study is the assumption that diagnostic sensitivity and specificity for a single diagnostic test do not vary substantially among settings (implied by estimating a single posterior for specificity and sensitivity using data from multiple surveys that used the same diagnostic). In reality, there may be some variation in diagnostic performance between settings, particularly for HTT serological diagnostics. For example, the widely used copro-Ag ELISA for detecting adult tapeworm infection is not species specific (*Ng-Nguyen et al., 2017*). Literature estimates indicate high specificity of HCC diagnostics; however, a proportion of positive results by both HCC Ag- and Ab- diagnostics may be due to transient responses and cross-reactions following exposure to the eggs of other other helminths including *Taenia saginata*, *Echinoccous granulosus* and *Schistosomia species* (*Lightowlers et al., 2016b*; *Noh et al., 2014*; *Kojic and White, 2003*; *Furrows et al., 2006*). Limited information

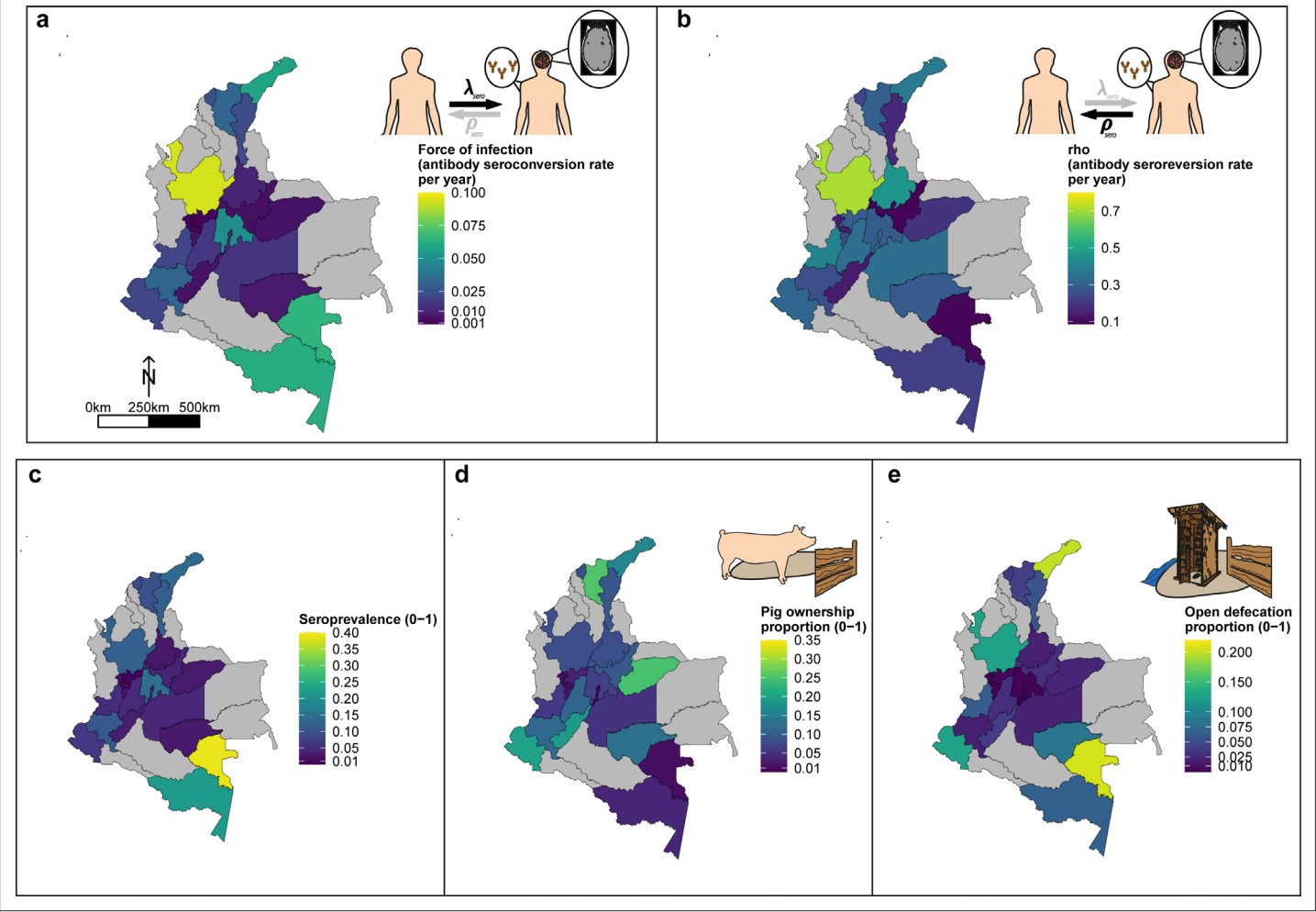

**Figure 7.** Geographic variation in, (**a**) HCC antibody (Ab)-seroconversion rate ($\lambda_{sero}$ or FoI), and (**b**) HCC Ab-seroreversion rate ($\rho_{sero}$), (**c**) human cysticercosis Ab all-age seroprevalence data, (**d**) pig ownership proportion and (**e**) open defecation reported proportion by department. The FoI ($\lambda_{sero}$) and Ab-seroreversion ($\rho_{sero}$) rates are parameter estimates obtained from the best-fit model with HCC Ab-seroconversion and Ab-seroreversion (reversible model). Note that San Andrés department is not clearly shown because of its size (small islands located in the top-left of a-c maps). For context, an all-age HCC Ab-seroprevalence=0.126, HCC Ab-seroconversion rate (FoI or $\lambda_{sero}$)=0.023 year-1, and HCC *Ab*-seroreversion rate ($\rho_{sero}$)=0.19 year-1 was obtained in San Andrés.

The online version of this article includes the following figure supplement(s) for figure 7:

**Figure supplement 1.** Geographic variation in, (**a**) proportion of pigs which free-roam as a management strategy and (**b**) proportion of pigs which free roam or are held in mixed systems as a management strategy.

exists on the co-distribution and prevalence of potentially cross-reactive *Taenia* species (such as *Taenia saginata*) and other helminths to determine the relative contribution of within- and between-location variability in the performance of specific diagnostic tests.

The prevailing (and most parsimonious) assumption governing the behaviour of the catalytic models fitted in this analysis is of a constant FoI with respect to age. Collated age-(sero)prevalence profiles presented here largely reveal increasing (sero)prevalence with age, or saturation with age through Ab-seroreversion/infection loss, but this behaviour can also be observed with age-dependent infection rates (*Grenfell and Anderson, 1985*). However, the available data limit the testing of more complex FoI functions. A few datasets appeared indeed to deviate from the basic assumption, for example for HTT dynamics in the DRC (*Madinga et al., 2017*) and Bolívar in Colombia (*Flórez Sánchez et al., 2013*). While it was not possible to fit the current formulation of catalytic models to these datasets, the specific age-prevalence peaks in younger ages may indicate other drivers such as heterogeneity in exposure (by age) or age-related immunity mechanisms. The observations of HTT copro-Ag-ELISA

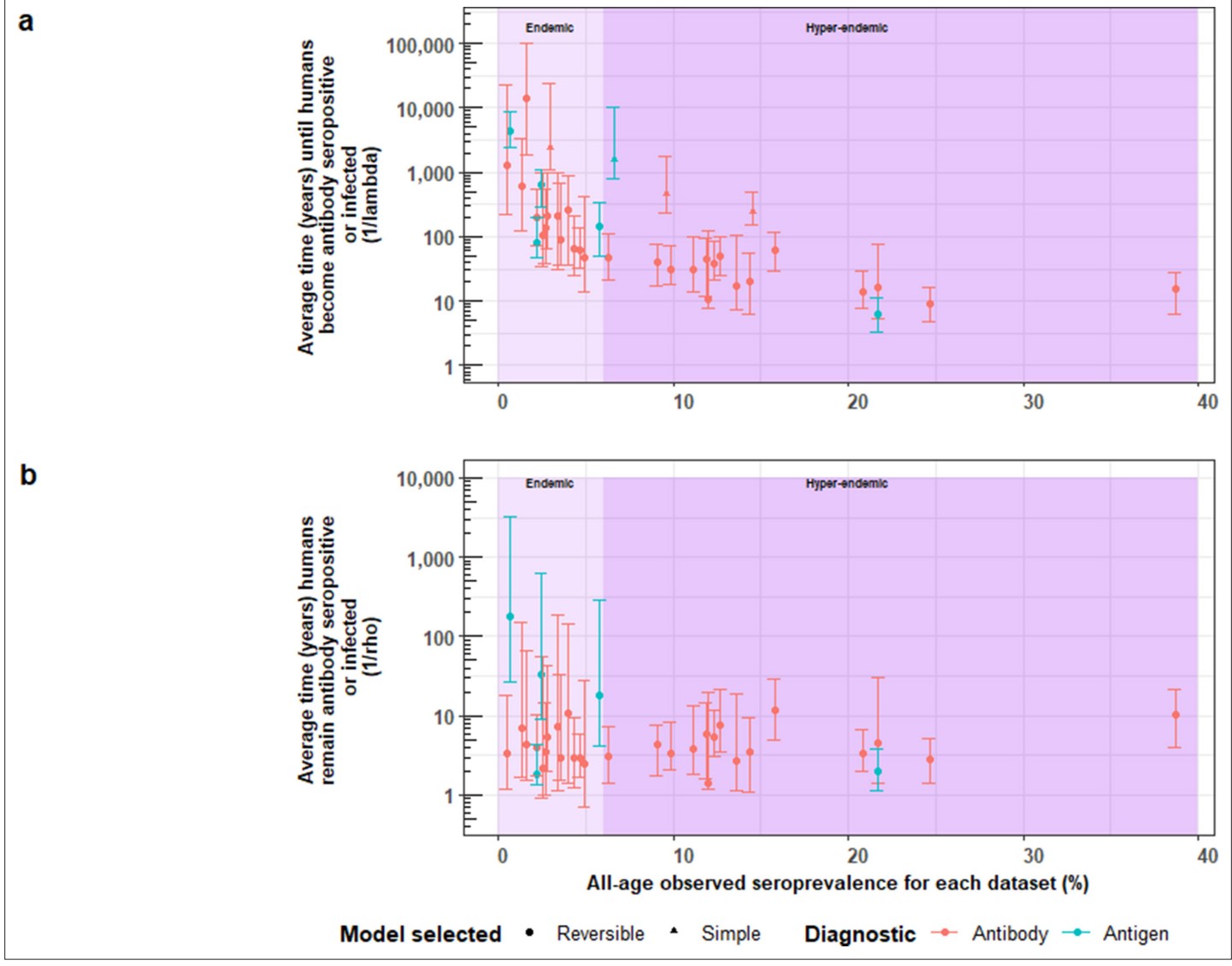

**Figure 8.** The relationship between (**a**) the average time until humans become cysticercosis antibody (Ab)-seropositive or infected ($1/\lambda$) and overall (all-age) seroprevalence, and (**b**) the average time humans remain cysticercosis Ab seropositive or infected ($1/\rho$) and overall (all-age) seroprevalence. The plot is stratified by proposed endemicity levels defined as endemic (>0% and <6% all-age HCC seroprevalence), and hyperendemic (≥6% all-age HCC seroprevalence).

The online version of this article includes the following figure supplement(s) for figure 8:

**Figure supplement 1.** Country-specific estimates of (**a**) the average time (in years) until humans become cysticercosis antibody (Ab)-seropositive or infected ($1/\lambda$, vertical axis), and (**b**) the average time (in years) humans remain HCC Ab-seropositive or infected ($1/\rho$, vertical axis).

**Figure supplement 2.** Trends in $\rho$ for increasing force-of-infection (FoI; $\lambda$) estimates for antigen-based estimates (**a**) and antibody (Ab)-based estimates (**b**), and trends in the infection loss ($\rho_{inf}$) to infection acquisition ($\lambda_{inf}$) rates by increasing prevalence for antigen-based estimates (**c**) and trends in Ab-seroreversion ($\rho_{sero}$) to Ab-seroconversion ($\lambda_{sero}$) rates by increasing Ab-seroprevalence for Ab-based estimates (**d**).

age-prevalence peaks/odds of infection in children from Peru, Guatemala and Zambia (*Falcon et al., 2003*; *Allan et al., 1996*; *Mwape et al., 2015*) suggest that these trends might be more widespread. Age peaks in HTT prevalence could indicate the need for targeted mass drug administration (MDA) programmes, such as delivery of anthelminthics (praziquantel) to school-age children, an approach that could be integrated into existing schistosomiasis control programmes where HTT and schistosomiasis are co-endemic. Should more age-prevalence data become available, especially for HTT, fitting models with age-varying FoI may become relevant as indicated in other NTDs (*Gambhir et al., 2009*).

The results presented here, particularly for HCC, where many datasets were available, suggest marked geographical heterogeneity in FoI and associated all-age (sero)prevalence estimates. This indicates substantial variation in the intensity of *T. solium* transmission among endemic settings. There is strong evidence for both HCC Ab-seroreversion and infection loss, likely due to transient dynamics which highlights the need for careful interpretation of cross-sectional (sero)prevalence survey estimates. Catalytic models provide useful tools for interpreting such data, which are far more abundant than a few longitudinal surveys reported in the literature. We also quantify, for the first time, the presence of substantial subnational variation in exposure to HCC, highlighting the need for tailored, subnational control strategies. More work is also required to understand whether age-prevalence peaks (as observed for HTT) are more commonplace, and whether age-targeted control strategies may be required under specific epidemiological and socioeconomic conditions.

## Methods

### Identifying relevant literature, data sources and data extraction

Published articles with HTT and HCC age-(sero)prevalence *or* age-infection data suitable for constructing age-stratified profiles were identified through two routes. Firstly, by extracting eligible studies from a systematic review of *T. solium* HTT and HCC (sero)prevalence global ranges (*Coral-Almeida et al., 2015*), and secondly, by updating the *Coral-Almeida et al., 2015* review, using the same strategy (see Appendix 1), for the period 01/11/2014 to 02/10/2019.

### Sub-national dataset for Colombia

*Flórez Sánchez et al., 2013* conducted a country-level survey of HCC antibody seroprevalence across 24 of Colombia's 32 administrative departments, sampling 29,360 individuals. Permission was granted from *Instituto National de Salud* to analyse these data to construct age-seroprevalence profiles. The study collected human cysticercosis Ab-seroprevalence data in the period 2008–2010,, alongside risk-factor data using a three-stage clustered random sampling framework. To maintain suitable sample sizes for model fitting, age-seroprevalence profiles were constructed at the department level (n=850–1291; *Figure 6—figure supplement 2*), rather than at municipality level (n=40–1140). Age-seroprevalence profiles stratified by sex in each department revealed no clear differences (*Figure 6—figure supplement 3*), therefore age-seroprevalence data were not stratified by sex for further analysis (but see *Galipó et al., 2021*). Due to difficulty fitting the specific catalytic models to data from the department of Bolívar, only 23 of 24 departments were assessed (n=28,100).

### Force-of-infection modelling for human taeniasis and human cysticercosis

The FoI, which describes the average (per capita) rate at which susceptible individuals seroconvert (become Ab-positive) or become infected, was estimated for HTT and HCC for each dataset. Simple and reversible catalytic models (*Figure 2*), originally proposed for fitting to epidemiological data by *Muench, 1934*, were fitted to HTT and HCC age-(sero)prevalence profiles. Model parameters fitted to Ab-datasets were interpreted as Ab-seroconversion ($\lambda_{sero}$) and Ab-seroreversion ($\rho_{sero}$) rates, while (copro-) Ag-datasets were interpreted as infection acquisition ($\lambda_{inf}$) and infection loss ($\rho_{inf}$) rates (*Figure 2* provides further details of model structure and parameter interpretation). Like the FoI, the infection loss/ seroreversion parameter ($\rho$) was also permitted to vary across settings (*Dermauw et al., 2018*). These two model configurations represent fundamentally different processes (i.e. presence or absence of Ab-seroreversion or infection loss). Therefore, it was the intention of this analysis to explore which of these processes best captures available age-(sero)prevalence data across different epidemiological settings, given the current knowledge gaps in the literature. In particular, there is minimal knowledge relating to how long taeniasis and cysticercosis infections and seropositivity persist in human hosts, on average (estimated here by the reciprocal of infection loss and Ab-seroreversion rates). This enabled us to consider the possibility that transmission intensity drives the presence and extent of infection loss and Ab-seroreversion rates (due to transient antigen responses and/or partial parasite establishment, as explored in the Discussion).

The true prevalence *p(a)* at age *a* is given for the simple model by,

$$p\left(a\right) = 1 - e^{-\lambda(a)} \tag{1}$$

and for the reversible model by,

$$p\left(a\right) = \frac{\lambda}{\lambda+\rho}\left[1 - e^{-(\lambda+\rho)(a)}\right] \tag{2}$$

## Model fitting and comparison

Analyses were performed in (*R Development Core Team, 2021*), following *Dixon et al., 2020*. A likelihood was constructed assuming that the observed data (representing a binary presence/absence of markers related to exposure or infection) are a realization of an underlying binomial distribution with probability p(a), given by the catalytic model, and adjusted to give the observed prevalence, p'(a), by the sensitivity (se) and specificity (sp) of the diagnostic adopted in the respective datasets (*Diggle, 2011*),

$$p'\left(a\right) = \left(1 - sp\right) + \left(se + sp\right) - 1 \times p\left(a\right) \tag{3}$$

Therefore, the likelihood of the data on the number of observed Ab-seropositive or infected humans of age a, r(a), from n(a) humans is,

$$L(r, n|\theta) = \Pi_a p'(a)^{r(a)}[1 - p'(a)]^{n(a)-r(a)} \tag{4}$$

where θ denotes, generically, the catalytic model (i.e. FoI, seroreversion/infection loss) and diagnostic performance (i.e. sensitivity and specificity) parameters. When the same diagnostic was used across surveys, sensitivity and specificity were assumed to be the same among surveys, yielding a single posterior distribution for the diagnostic performance (sensitivity, specificity) for each test (whilst retaining dataset-specific estimates of $\lambda$ and $\rho$). The approach captures uncertainty in diagnostic sensitivity and specificity, but does not permit variation in performance across settings.

A Bayesian Markov chain Monte Carlo (MCMC) Metropolis–Hastings sampling algorithm was implemented to estimate the parameter posterior distribution f(θ|r, n), assuming uniform prior distributions for $\lambda$ and $\rho$ (*Table 4*). A weakly informative prior for $\lambda$, assuming a lognormal distribution (mean informed by the median estimate from the simple model fit, and a standard deviation of 1), was used for reversible model fits to prevent $\lambda$ chains drifting to flat posterior space and failing to converge.

Informative beta distribution priors for the diagnostic sensitivity and specificity were defined to capture literature estimates of the mean and 95%CIs for these parameters; *Figure 9* and *Figure 10* with *Supplementary file 1*, provide more detail.

A maximum of 20,000,000 iterations were run for models fitted simultaneously to multiple datasets (2,000,000 for models fitted to single datasets) to obtain a sufficient sample to reduce autocorrelation through substantial subsampling, with the first 25% of runs being discarded as burn-in. The parameter posterior distributions, used to generate fitted prevalence curves and associated uncertainties for each model fit, were summarised using the median and 95% Bayesian credible intervals (95% BCIs).

**Table 4.** Range for force-of-infection (FoI), $\lambda$, and seroreversion / infection loss, $\rho$, uniform priors.

| **T.** T. solium indicator | $\lambda$: limits on uniform prior (year$^{-1}$) | $\rho$: limits on uniform prior (year$^{-1}$) | Rationale |
|---|---|---|---|
| HTT (copro-antigen (Ag) and antibody (Ab)) | Minimum: 0 Maximum: 12 | Minimum: 0.333 Maximum: 1 | $\lambda$: maximum seroconversion rate of 12 year$^{-1}$=1 month$^{-1}$ (1/$\lambda$), representing the lower limit before humans become copro-Ag positive (i.e., infected) or Ab-seropositive of 1 month $\rho$: minimum Ab-seroreversion or infection loss rate of 0.333 year$^{-1}$=0.0277 month$^{-1}$, or 36 month duration (1/$\rho$) for humans to remain copro-Ag (i.e. infected) or Ab- seropositive, reflecting the upper average limit assumed for the life- expectancy of adult *Taenia solium* of 3 years (*Gonzalez and Rosenheck, 2002*); a maximum of 0.0833 month$^{-1}$ represents the minimum time humans remain copro-Ag (i.e., infected) or Ab-seropositive reflecting the minimum life-expectancy of the adult worm of 12 months (*García et al., 2003*) (1/$\rho$) |
| HCC (antigen and antibody) | Minimum: 0 Maximum: 12 | Minimum: 0 Maximum: 12 | Both $\lambda$ and $\rho$: maximum infection acquisition/loss or Ab-seroconversion / Ab-seroreversion rate of 12 year$^{-1}$=1 month$^{-1}$ represents the lower limit before humans acquire infection or become Ab-seropositive (1/$\lambda$), or remain HCC antigen (i.e., infected) or Ab-seropositivity (1/$\rho$) of 1 month. |

N.B. The reciprocal of the rates of $\lambda$ and $\rho$ give the duration of susceptibility (1/$\lambda$; 1/$\lambda$ sero for Ab-based, or 1/$\lambda$ inf for Ag-based) or of remaining infected or Ab-seropositive (1/$\rho$; 1/$\rho$ sero for Ab-based, or 1/$\rho$ inf for Ag-based). HTT: human taeniasis; Ab: antibody; Ag: antigen; HCC: human cysticercosis.

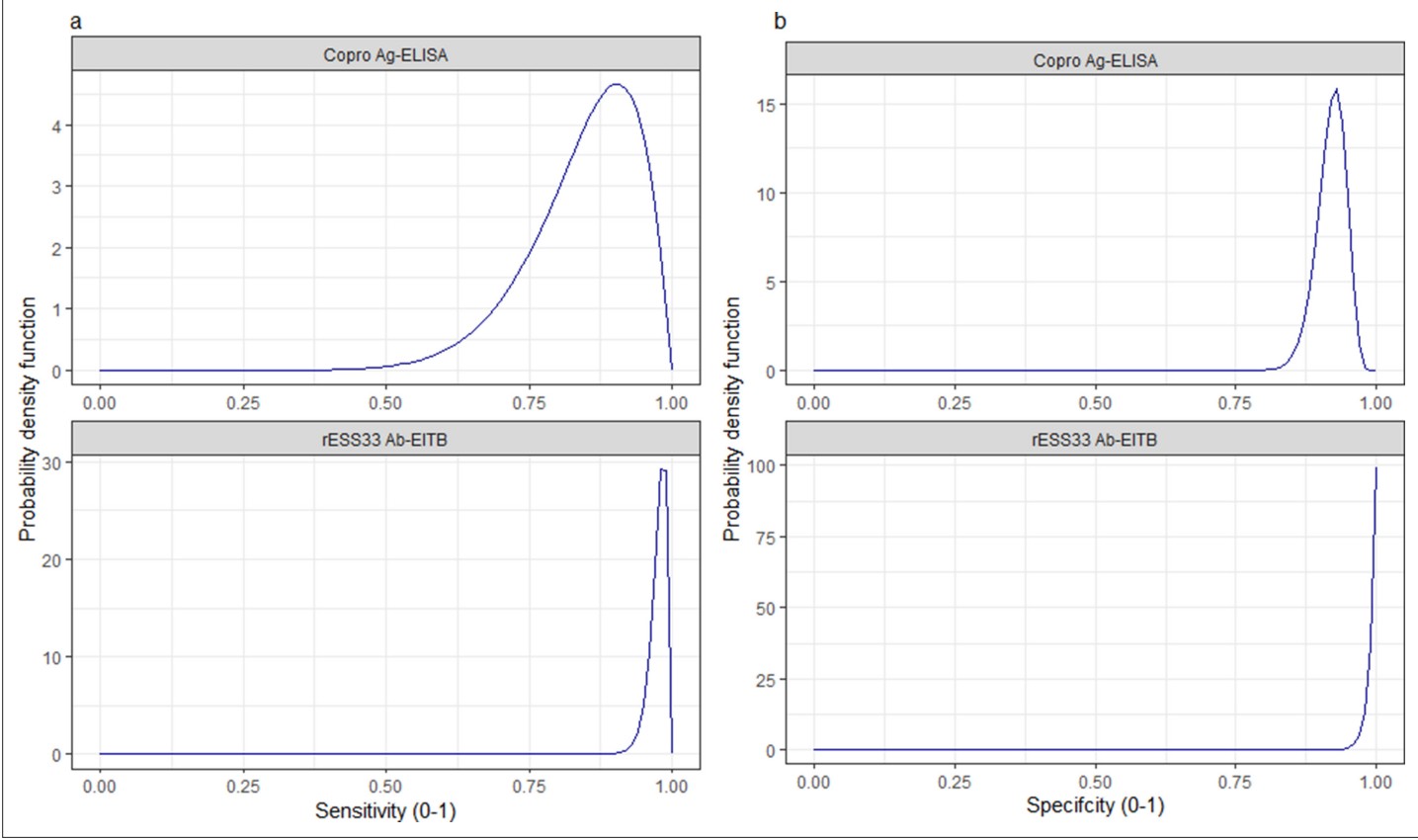

**Figure 9.** Informative beta distribution priors constructed for sensitivity and specificity parameters for each human taeniasis diagnostic. $\beta$ prior distributions for: (**a**) sensitivity (se) (left-hand column plots); (**b**) specificity (sp) (right-hand column plots) of each diagnostic, constructed with $\alpha$ and $\beta$ shape parameters provided in *Supplementary file 1*. The $\beta$ distribution provides a more flexible alternative to the uniform distribution, where the parameters of interest are constrained between 0 and 1. The shape parameters were chosen based on the literature estimates of se and sp (whereby $\alpha/(\alpha+\beta)$) equals the mean of the distribution (*Lambert, 2018*), and adapted where required to ensure that the spread of the prior distribution included the 95% confidence intervals identified in the literature. Ag: antigen; ELISA: enzyme-linked immunosorbent assay; Ab: antibody; EITB: enzyme-linked immunoelectrotransfer blot.

Simple and reversible model fits, after being fitted to all datasets, were compared using the deviance information criterion (DIC) (*Spiegelhalter et al., 2002*), with the model yielding the smallest DIC score being selected (*Supplementary files 3–5*).

## Data and code availability

Code to replicate the analysis is provided at https://github.com/mrc-ide/human_tsol_FoI_modelling, (copy archived at swh:1:rev:b3eb4c05a42a882c13fb755c3e50dfda0f3e4ef3; *Dixon, 2022*). All age-(sero)prevalence data are available in the following data repository: http://doi.org/10.14469/hpc/10047. Original data for two datasets available (under the Creative Commons Attribution License; CC BY 4.0) from the International Livestock Research Institute open-access repository (http://data.ilri.org/portal/dataset/ecozd) referenced in *Holt et al., 2016* and University of Liverpool open-access repository (http://datacat.liverpool.ac.uk/352/) referenced in *Fèvre et al., 2017*.

## Acknowledgements

MAD, PW, CW, ZMC and MGB acknowledge funding from the Medical Research Council (MRC) Centre for Global Infectious Disease Analysis (reference MR/R015600/1), jointly funded by the UK MRC and the UK Foreign, Commonwealth & Development Office (FCDO), under the MRC/FCDO Concordat agreement and is also part of the European and Developing Countries Clinical Trials Partnership (EDCTP2) programme supported by the European Union.

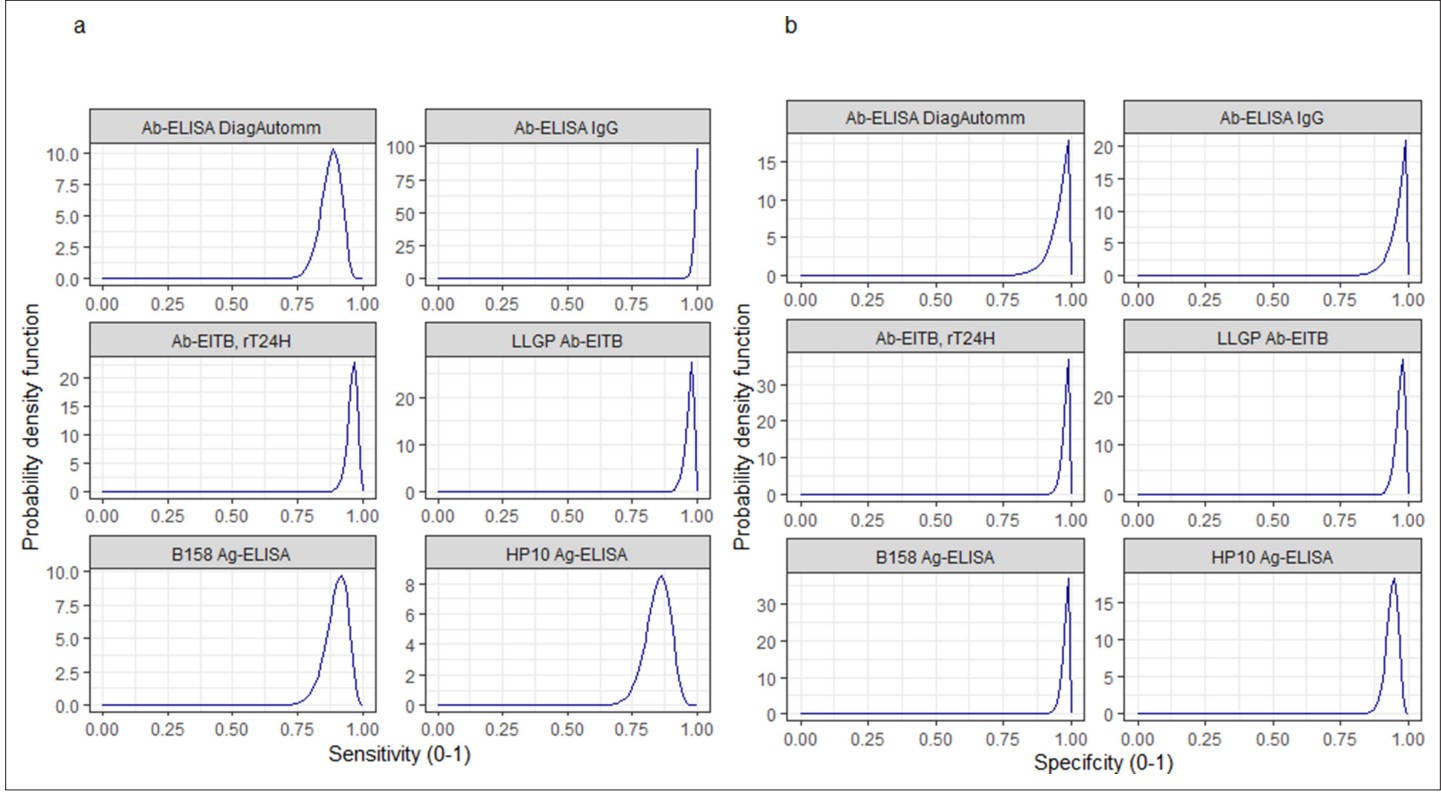

**Figure 10.** Informative beta distribution priors constructed for sensitivity and specificity parameters for each human cysticercosis diagnostic. $\beta$ prior distributions for: (**a**) sensitivity (se) and (**b**) specificity (sp) of each diagnostic, constructed with $\alpha$ and $\beta$ shape parameters provided in **Supplementary file 1**. The $\beta$ distribution provides a more flexible alternative to the uniform distribution where the parameters of interest are constrained between 0 and 1 (**Lambert, 2018**). The shape parameters were chosen based on the literature estimates of se and sp (whereby $\alpha/(\alpha+\beta)$) equals the mean of the distribution (**Lambert, 2018**), and adapted where required to ensure that the spread of the prior distribution included the 95% confidence intervals identified in the literature. Ab: antibody; ELISA: enzyme-linked immunosorbent assay; EITB: enzyme-linked immunoelectrotransfer blot; LLGP: lentil lectin-purified glycoprotein; Ig: immunoglobulin; Ag: antigen.

# Additional information

### Funding

| Funder | Grant reference number | Author |
|---|---|---|
| Medical Research Council | MR/R015600/1 | Matthew A Dixon |

The funders had no role in study design, data collection and interpretation, or the decision to submit the work for publication.

### Author contributions

Matthew A Dixon, Conceptualization, Data curation, Software, Formal analysis, Funding acquisition, Validation, Investigation, Visualization, Methodology, Writing – original draft, Writing – review and editing; Peter Winskill, Formal analysis, Supervision, Methodology, Writing – review and editing; Wendy E Harrison, Supervision, Writing – review and editing; Charles Whittaker, Software, Methodology, Writing – review and editing; Veronika Schmidt, Formal analysis, Writing – review and editing; Astrid Carolina Flórez Sánchez, Agnes U Edia-Asuke, Data curation, Writing – review and editing; Zulma M Cucunuba, Data curation, Methodology, Writing – review and editing; Martin Walker, Formal analysis, Methodology, Writing – review and editing; María-Gloria Basáñez, Formal analysis, Supervision, Methodology, Writing – original draft, Writing – review and editing

### Author ORCIDs

Matthew A Dixon http://orcid.org/0000-0002-1710-6237

Zulma M Cucunuba http://orcid.org/0000-0002-8165-3198

**Decision letter and Author response**
Decision letter https://doi.org/10.7554/eLife.76988.sa1
Author response https://doi.org/10.7554/eLife.76988.sa2

## Additional files

### Supplementary files

• Supplementary file 1. Summary of studies included in final analysis and the diagnostic parameters used to set the probabilistic constraints for sensitivity and specificity of each test. [†]Sensitivity and specificity estimated in a Bayesian framework by *Praet et al., 2013*; [††] sensitivity and specificity based on exposure estimated in a Bayesian framework by *Praet et al., 2010*; *limited information on the specific protocol used for the copro-antigen assay; * *Flórez Sánchez et al., 2013* in*dicate that this diagnostic is suitable to determine exposure through detection of anti-cysticercus immunoglobulin G (IgG) antibodies and was evaluated in Colombian patients to assess cross-reactions with different infectious agents (Taenia saginata, Hymenolepis nana, Echinococcus sp., Fasciola hepatica, Entamoeba histolytica, Ascaris lumbricoides, Mansonella ozzardi, Treponema pallidum, Cryptococcus neoformans and HIV)*; ** sensitivity and specificity estimated in a Bayesian framework by *Praet et al., 2010*; [‡] 95%CIs not provided to inform priors, therefore minimal uncertainty introduced around the sensitivity and specificity estimates to construct priors. Original data for two datasets available (under the Creative Commons Attribution License; CC BY 4.0) from the [‡‡]International Livestock Research Institute open-access repository (http://data.ilri.org/portal/dataset/ecozd) referenced in *Holt et al., 2016* and [‡‡‡]University of Liverpool open-access repository (http://datacat.liverpool.ac.uk/352/) referenced in *Fèvre et al., 2017*; [ı] sensitivity and specificity estimated in *Fleury et al., 2007*. DRC: Democratic Republic of the Congo; LPDR: Lao People's Democratic Republic; Ag; antigen; Ab: antibody; ELISA: enzyme-linked immunosorbent assay; LLGP-EITB: Lentil lectin-purified glycoprotein enzyme-linked immunoelectrotransfer blot; Ig: immunoglobulin; km: kilometers, PCC: porcine cysticercosis; RCT: randomized controlled trial.

• Supplementary file 2. Observed prevalence estimates from sub-Saharan Africa, South America, and Asia for studies referring to "hyper-"or "highly-"endemic setting for human taeniasis (HTT) and human cysticercosis (HCC). * rES38 Ab-immunblot (*Wilkins et al., 1999*) ** rT24H Ab-ELISA (*Hancock et al., 2006*) [†]B158/B60 Ag-ELISA (*Brandt et al., 1992*; *Dorny et al., 2000*) [††] LLGP-EITB (*Tsang et al., 1989*) [ı]Ab-ELISA confirmed by immunoblot (*Sato et al., 2018*) [ıı]Copro-PCR (*Sato et al., 2018*). DRC: Democratic Republic of the Congo (DRC); Lao PDR: Lao People's Democratic Republic.

• Supplementary file 3. The deviance information criterion (DIC) and parameter estimates for simple and reversible catalytic models fitted to each observed human taeniasis (antibody and antigen) age-seroprevalence dataset (ordered by decreasing value of all-age seroprevalence). For diagnostic methods used see the corresponding study in *Supplementary file 1*. Seroprevalence results are accompanied by 95% confidence intervals (95% CI) calculated by the Clopper-Pearson exact method. Parameter median posterior estimates are presented with 95% Bayesian credible intervals (95% BCI) and Deviance information criterion (DIC) model fitting scores; * Diagnostic sensitivity and specificity jointly fitted for the Copro-Ag ELISA (*Coral-Almeida et al., 2015*). [†] Best-fitting model determined by DIC (jointly-fitted dataset). [††] Best-fitting model determined by DIC (individually-fitted dataset). NA = Not applicable; PDR: People's Democratic Republic.

• Supplementary file 4. The deviance information criterion (DIC) and parameter estimates for simple and reversible catalytic models fitted to each observed human cysticercosis antibody age-seroprevalence dataset (ordered by decreasing value of all-age seroprevalence). For diagnostic methods used see the corresponding study in *Supplementary file 1*. Seroprevalence results are accompanied by 95% confidence intervals (95% CI) calculated by the Clopper-Pearson exact method. Parameter median posterior estimates are presented with 95% Bayesian credible intervals (95% BCI) and Deviance information criterion (DIC) model fitting scores; * Diagnostic sensitivity and specificity for the antibody lentil lectin-purified glycoprotein enzyme-linked immunoelectrotransfer blot (Ab LLGP-EITB) assay (*Tsang et al., 1989*) were jointly fitted across datasets. ** Diagnostic sensitivity and specificity for the antibody Ab-ELISA (DiagnosticAutomation/CortezDiagnostic (2006)) assay were jointly fitted across datasets. [†] Best-fitting model determined by DIC (jointly-fitted dataset). [††] Best-fitting model determined by DIC (individually-fitted dataset). NA = Not applicable; PDR: People's Democratic Republic.

• Supplementary file 5. The deviance information criterion (DIC) and parameter estimates for simple and reversible catalytic models fitted to each observed human cysticercosis antigen age-seroprevalence dataset (ordered by decreasing value of all-age seroprevalence). For diagnostic methods used see the corresponding study in *Supplementary file 1*. Seroprevalence results are accompanied by 95% confidence intervals (95% CI) calculated by the Clopper-Pearson exact method. Parameter median posterior estimates are presented with 95% Bayesian credible intervals (95% BCI) and Deviance information criterion (DIC) model fitting scores; * Diagnostic sensitivity and specificity for the B158/B60 Ag-ELISA (*Brandt et al., 1992*; *Dorny et al., 2000*). † Best-fitting model determined by DIC (jointly-fitted dataset). †† Best-fitting model determined by DIC (individually-fitted dataset). NA = Not applicable; DRC: Democratic Republic of the Congo; PDR: People's Democratic Republic.

• Supplementary file 6. The deviance information criterion (DIC) and parameter estimates for the reversible catalytic model jointly fitted (for diagnostic sensitivity and specificity) to the observed human cysticercosis antibody age-seroprevalence for each available department in Colombia (n=23, ordered by decreasing value of all-age seroprevalence). For diagnostic methods used see the corresponding study in *Supplementary file 1*. DIC score for the reversible model was –401.78. Jointly-fitted diagnostic sensitivity was 0.989 (95%BCI: 0.946–0.999) and specificity was 0.998 (95%BCI: 0.993–0.999). Seroprevalence results are accompanied by 95% confidence intervals (95% CI) calculated by the Clopper-Pearson exact method. Parameter median posterior estimates are presented with 95% Bayesian credible intervals (95% BCI) and Deviance information criterion (DIC) model fitting scores.

• Supplementary file 7. The deviance information criterion (DIC) and parameter estimates for the simple catalytic model jointly fitted (for diagnostic sensitivity and specificity) to the observed human cysticercosis antibody age-seroprevalence for each available department in Colombia (n=23, ordered by decreasing value of all-age seroprevalence). For diagnostic methods used see the corresponding study in *Supplementary file 1*. DIC score for the simple model was –442.7. Jointly-fitted diagnostic sensitivity was 0.987 (95%BCI: 0.966–0.999) and specificity was 0.980 (95%BCI: 0.975–0.984). Seroprevalence results are accompanied by 95% confidence intervals (95% CI) calculated by the Clopper-Pearson exact method. Parameter median posterior estimates are presented with 95% Bayesian credible intervals (95% BCI) and Deviance information criterion (DIC) model fitting scores.

• Supplementary file 8. PRIME-NTD (Policy-Relevant Items for Reporting Models in Epidemiology of Neglected Tropical Diseases) Summary. Adapted from *Behrend et al., 2020*. Full formulation of the principles: Don't do it alone. Engage stakeholders throughout, from the formulation of questions to the discussions on the implications of the findings. Reproducibility is key! Prepare and make available (preferably open-source) a complete technical documentation of all model code, mathematical formulas, assumptions and their justification, allowing others to reproduce the model. Model calibration, goodness-of-fit and validation are fundamental processes of scientific modelling. All data used should be described in sufficient detail to allow the reader to assess the type and quality of these analyses. When using data by reference, use Principle 2. Communicating uncertainty is a hallmark of good modelling practice. Perform a sensitivity analysis of all key parameters, and for each paper reporting model predictions include an uncertainty assessment of those model outputs within the paper. Model outcomes should be articulated in the form of testable hypotheses. This allows comparison with other models and future events as part of the ongoing cycle of model improvement. Incorporating prior work by reference in the paper is sufficient. Write what relevant information is contained in the prior work; then note its reference number in the summary table. Please verify that the whole chain of steps in referenced work is actually complete and up-to-date.

• Transparent reporting form

### Data availability

All age-(sero)prevalence data are available in the following data repository: http://doi.org/10.14469/hpc/10047. Original data for two datasets available (under the Creative Commons Attribution License; CC BY 4.0) from the International Livestock Research Institute open-access repository (http://data.ilri.org/portal/dataset/ecozd) referenced in Holt et al. (2016) and University of Liverpool open-access repository (http://dx.doi.org/10.17638/datacat.liverpool.ac.uk/352) referenced in Fèvre et al. (2017).

The following previously published datasets were used:

| Author(s) | Year | Dataset title | Dataset URL | Database and Identifier |
|---|---|---|---|---|
| Holt HR, Inthavong P, Khamlome B, Blaszak K, Keokamphe C, Somoulay V, Phongmany A, Burr PA, Graham K, Allen J, Donnelly B, Blacksell SD, Unger F, Grace D, Alonso S, Gilbert J | 2016 | Ecosystem approaches to the better management of zoonotic emerging infectious diseases in the Southeast Asia region | http://data.ilri.org/portal/dataset/ecozd | ILRI Data Portal, ecozd |
| Fèvre EM, de Glanville WA, Thomas LF, Cook EAJ, Kariuki S, Wamae CN | 2017 | An integrated study of human and animal infectious disease in the Lake Victoria Crescent small-holder crop-livestock production system, Kenya | http://datacat.liverpool.ac.uk/352/ | DataCat: The Research Data Catalogue, 10.17638/datacat.liverpool.ac.uk/352 |

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

## Appendix 1

### Systematic literature search strategy

Literature searches were conducted for the period 01/11/2014 to 29/01/2019, according to the following outlined in *Coral-Almeida et al., 2015*:

In PubMed, using the Boolean operator AND, the terms "cysticercosis" AND "Taenia solium" AND "epidemiology" were introduced in the main search bar and the filters were activated for the period from 1988/12/31 to 2014/10/31.

In Web of Science, the strategy applied was introducing in the basic search bar the topic "cysticercosis" adding fields with the correspondent Boolean operators: AND Topic = (*Taenia solium*) AND Topic = (epidemiology).

In the Latin American & Caribbean Health Sciences Literature (LILACS) database, the strategy adopted consisted in introducing in the main search bar the terms "cysticercosis Taenia solium". In the latter case the term "epidemiology" was excluded to obtain the maximum return of articles since LILACS is a smaller targeted database when compared to PubMed and Web Of Science.

In addition the literature databases searched in *Coral-Almeida et al., 2015*, the African Journals Online (AJOL) was also searched, with the search strategy terms including "cysticercosis *Taenia solium*".

Following identification of article titles, the exclusion criteria at the title and abstract screening stage included: (1) wrong parasite species, (2) non-endemic countries, (3) only pig studies, (4) pre-clinical/clinical research only, (5) diagnostic development only, (6) non-epidemiological study/ primary data not collected, and (7) unrelated topic.

At the full-text screening stage, the following inclusion criteria were applied: (1) community-based studies with a cross-sectional or longitudinal study design assessing *T. solium* human taeniasis (HTT).

### Defining endemicity thresholds

Observed prevalence estimates from studies mentioning "hyper" or "highly" endemic in the literature are reported in *Supplementary file 2*. We define the threshold between endemic and hyperendemic levels based on the lowest value for human taeniasis prevalence (2.8%, rounded to 3%) and human cysticercosis prevalence (5.8%, rounded to 6%), with studies reporting observed prevalence estimates above this threshold representing hyperendemic settings.

