## [Editor Report]

Dixon and colleagues have collated published "age-prevalence" data from 16 studies (4 from South America, 8 from Africa, and 4 from Asia) to estimate the force of infection of taeniasis/human cysticercosis across diverse endemic settings. This study addresses a major knowledge gap, as little is currently known regarding the extent of current/recent Taenia solium transmission worldwide. The main limitation of this interesting study originates from the very nature of the primary data analyzed. Authors examine how the prevalence of (genus- but not species-specific) antigens shed in the stools or in the plasma and (more or less stage- and species-specific) antibodies changes across age groups and fit simple catalytic models to these very heterogeneous datasets.

---

## [Decision Letter]

**Decision letter after peer review:**

Thank you for submitting your article "Global Force-of-Infection Trends for Human *Taenia solium* Taeniasis/Cysticercosis" for consideration by *eLife*. Your article has been reviewed by 3 peer reviewers, including Marcelo U Ferreira as Reviewing Editor and Reviewer #1, and the evaluation has been overseen by Miles Davenport as the Senior Editor.

Essential revisions:

1) Please add a more detailed discussion about the limitations of the dataset analyzed (reviewer #1). Address the potential differences between antigen and antibody detection, the differences between laboratory methods used in each primary study (e.g., species-specificity, sensitivity) and openly discuss how these differences may affect the current analysis.

2) Please specifically address why the reversible catalytic model has been used to fit some data, while the simple (non-reversible) version has been used to fit other datasets. What is the rationale for using such models with such different assumptions (reviewer #2)?

3) Please correct the typos listed by reviewers #2 and #3.

*Reviewer #1 (Recommendations for the authors):*

Some suggestions are listed below. Given that no new data are to be generated by the authors, I primarily focus on ways of acknowledging the limitations of available data and discussing how can they impact this study.

1. I would like to see a much more comprehensive discussion on "seroreversion" in human taeniasis and cysticercosis. The basic question is: how quickly antibodies become undetectable after treatment or following spontaneous worm death and elimination? Do seroreversion rates depend on average levels of antibodies at baseline, before treatment (i.e., are individuals heavily exposed to infection within the past few years more likely to remain seropositive long after treatment even in the absence of reinfection)?

2. If antibodies are not quickly cleared, then seroconversion rates are not synonymous with "infection rates". In a population with heterogeneous exposure to infection, individuals more frequent exposed to infection may remain seropositive for longer (even if treated) because of memory responses and their antibody status will not change upon reinfection (i.e., they will remain positive). There is a potential bias towards underestimating infection rates from seroconversion data -- especially among individuals who are most likely to be (re)infected..

3. If the antibody-detection assay used in the Colombian study (described in a local publication, not available to this reviewer) is not strictly species- and stage-specific, and also short-lived (i.e., if only individuals currently exposed to T. solium cysticerci are positive), the age-related prevalence of "cysticercosis" may have been grossly overestimated in this country.

4. If prevalence data from selected study sites in a dozen countries cannot be taken as representative of the countries/regions of origin, the study is far from being able to estimate "global force-of-infection trends". Please provide further information regarding the primary studies, in addition to those in Table S1, to help the reader decide whether or not some extrapolation is possible.

*Reviewer #2 (Recommendations for the authors):*

Although the work is well written and very organized, I have some concerns about the model and assumptions used in the analysis.

Equation 3 is different than the one reported by Diggle, 2011. One essential set of parentheses is missing. Please, clarify it.

The methods section would be improved by presenting the motivations for the models' choice and a brief description of them, including assumptions and limitations. Although the reversible catalytic model (rho>0) is a generalization of the simple form (rho=0), there are two mutually exclusive assumptions: either only conversion or conversion and reversion. Because of it, unless well justified, it seems unlike that a particular pattern of an infectious disease could be described considering two fundamentally different biological processes. As observed in Figures 2, 3 and 4, the data sometimes support the simple model, sometimes support the reversible, and sometimes it is not possible to decide. It means to me that it is not possible to decide between both designs which encapsulates the mechanism driving the disease age-prevalence presented by the data. Please, clarify it.

Another concern is regarding to the saturation of the reversible model. The current formulation assumes that all individuals are exposed/susceptible to the disease and, therefore, its saturation is controlled by the reversion rate parameter (rho). The greater rho, the lower the saturation. Hypothetically, one can think of a situation in which just a fraction of the population is exposed/susceptible to an underlined disease. In this scenario, the greater the fraction of individuals not exposed/susceptible, the greater the interference bias of the reversion rate. I wonder whether the authors believe that this is a reasonable possibility and, in the affirmative case, please acknowledge the limitation of the methodology.

---

## [Author Response]

Essential revisions:1) Please add a more detailed discussion about the limitations of the dataset analyzed (reviewer #1). Address the potential differences between antigen and antibody detection, the differences between laboratory methods used in each primary study (e.g., species-specificity, sensitivity) and openly discuss how these differences may affect the current analysis.

We thank Reviewer #1 and the Reviewing Editor for providing this feedback. A more detailed response to these points can be found in Author responses 1.3 and 1.8, where we focus on providing clarification on the diagnostic used in Flórez Sánchez *et al.,* (2013). We highlight the difference between inferences made from antibody-based surveys (relating to exposure) and antigen-based surveys (active infection dynamics) in Authors’ responses 1.2 and 1.7.

We also provide a detailed analysis of the limitations regarding current diagnostics and how these may impact our analysis on lines 411 – 415 regarding imperfect specificity, especially in low (sero)prevalence settings, and on lines 465 – 473 in the Discussion, focussing on implications regarding variation in sensitivity and specificity estimates across settings, as well as the presence of transient responses/ cross-reactions. In particular, our responses on diagnostic test limitations beyond those already mentioned can be found in Authors’ responses 1.1 in relation to taeniasis diagnostics.

2) Please specifically address why the reversible catalytic model has been used to fit some data, while the simple (non-reversible) version has been used to fit other datasets. What is the rationale for using such models with such different assumptions (reviewer #2)?

We thank Reviewer #2 and the Reviewing Editor for providing this feedback. We fitted both models (simple and reversible) to each dataset, to determine whether there is more evidence for biological processes including or excluding seroreversion / infection loss across settings, as this is a key knowledge gap in the field. The best-fit model was then selected and presented. A more detailed response to these points can be found under Authors’ response 2.2.

3) Please correct the typos listed by reviewers #2 and #3.

We thank Reviewers #2 and #3 and the Reviewing Editor for indicating these typographical errors. We will address the specific typos under our responses in the Reviewers #2 and #3 (Public Review and Recommendations) sections.

Reviewer #1 (Recommendations for the authors):Some suggestions are listed below. Given that no new data are to be generated by the authors, I primarily focus on ways of acknowledging the limitations of available data and discussing how can they impact this study.1. I would like to see a much more comprehensive discussion on "seroreversion" in human taeniasis and cysticercosis. The basic question is: how quickly antibodies become undetectable after treatment or following spontaneous worm death and elimination? Do seroreversion rates depend on average levels of antibodies at baseline, before treatment (i.e., are individuals heavily exposed to infection within the past few years more likely to remain seropositive long after treatment even in the absence of reinfection)?

We agree with the reviewer that this is an important area for further consideration. There is very limited literature available on the persistence of antibodies raised to the adult tapeworm after spontaneous worm death and elimination, and, as evidenced by the minimal number of available studies with suitable ageprevalence data, we were not able to estimate and assess human taeniasis seroreversion rates in different epidemiological settings. Thus, it is difficult to discuss these dynamics further in our paper at present. We therefore introduce further text to present these considerations as important further research questions that should form part of the T. solium research agenda:

**“**Understanding antibody kinetics in relation to spontaneous death and elimination of the adult tapeworm (either naturally or following treatment) is therefore an important area for further consideration, especially towards understanding whether seroreversion rates are related to baseline antibody levels.” (lines 396 – 399)

We have also included some further text, as indicated in Authors’ response 1.1, regarding challenges with measuring reinfection rates should antibodies to the adult tapeworm persist long after worm death.

We currently discuss, in detail, considerations around human cysticercosis (antibody and antigen) seroreversion rates, exploring possible underlying mechanisms to explain the presence/absence and variation in these rates through transient responses and partial establishment of cysts (in the paragraph on lines 416– 440).

2. If antibodies are not quickly cleared, then seroconversion rates are not synonymous with "infection rates". In a population with heterogeneous exposure to infection, individuals more frequent exposed to infection may remain seropositive for longer (even if treated) because of memory responses and their antibody status will not change upon reinfection (i.e., they will remain positive). There is a potential bias towards underestimating infection rates from seroconversion data -- especially among individuals who are most likely to be (re)infected.

The reviewer raises an important point. We have taken care not to conflate seroconversion rates with infection acquisition rates, relating the former to exposure and the latter to active infection dynamics. We have further clarified in the text that antibody seroconversion refers to exposure and not to active infection in lines 127 – 129, as follows:

“The FoI parameters of acquisition (λ) and reversion/loss (ρ) (for antibody datasets, a marker of exposure: λ_sero_ and ρ_sero_; for antigen datasets, a marker of active infection: λ_inf_ and ρ_inf_) were estimated for each dataset.”

We have also amended the title of Table 1 (see Authors’ response 1.2.), Table 2 and Table 3 to indicate that “Antibody seroconversion and seroreversion rates represent markers of exposure” and “Antigen-based infection acquisition and infection loss rates represent markers of active infection”.

3. If the antibody-detection assay used in the Colombian study (described in a local publication, not available to this reviewer) is not strictly species- and stage-specific, and also short-lived (i.e., if only individuals currently exposed to T. solium cysticerci are positive), the age-related prevalence of "cysticercosis" may have been grossly overestimated in this country.

We thank the reviewer for this question and have detailed in Authors’ responses 1.3 that the López et al., diagnostic used in the Flórez Sánchez et al., 2013 study is indicated as both highly stage- and species-specific (97.6% specific). Therefore, we do not believe that the age-prevalence is overestimated in this study. We agree with the reviewer that this information should be made available in the manuscript, therefore we include more details in the footnote section of Supplementary File 1:

“** Flórez Sánchez et al., (2013) indicate that this diagnostic is suitable to determine exposure through detection of anti-cysticercus immunoglobulin G (IgG) antibodies and was evaluated in Colombian patients to assess cross-reactions with different infectious agents (Taenia saginata, Hymenolepis nana, Echinococcus sp., Fasciola hepatica, Entamoeba histolytica, Ascaris lumbricoides, Mansonella ozzardi, Treponema pallidum, Cryptococcus neoformans and HIV)**”.**

For the benefit of the reviewer, we have also managed to locate a reference from 1996 describing the assay, which is downloadable from:

https://revistabiomedica.org/index.php/biomedica/article/view/903

Corredor et al., (1996) [Standardization and evaluation of ELISA in dried blood eluates collected on filter paper for the diagnosis of cysticercosis]. Biomedica 16**,** 131–133 (in Spanish).

4. If prevalence data from selected study sites in a dozen countries cannot be taken as representative of the countries/regions of origin, the study is far from being able to estimate "global force-of-infection trends". Please provide further information regarding the primary studies, in addition to those in Table S1, to help the reader decide whether or not some extrapolation is possible.

We agree and thank the reviewer for this feedback. We have addressed the issue of representativeness of each study (where information is available) under Authors’ response 1.4. In several studies, the authors indicated that the study sites, selected from different areas, were representative of specific socioeconomic factors across a region (e.g., Conlan *et al.,* 2012; Jayaraman *et al.,* 2011). We acknowledge that it was difficult to determine how representative some of the other studies were at country level. Having said that, we strongly contend that our analyses, taken in their entirety, do indicate substantial variation in FoI/seroreversion or infection loss estimates across a range of different epidemiological settings representing the major global endemic areas (e.g., South America, sub-Saharan Africa and Asia). The distribution of study sites is visually presented in Figure S2, which is now Figure 1—figure supplement 1: Geographical distribution of studies with human taeniasis (HTT) and human cysticercosis (HCC) age-(sero) prevalence data included in the final analysis (n = 16) by diagnostic method. For this reason, we believe this study reflects the variation in global trends, and therefore propose modifying the manuscript title as follows:

“Global variation in Force-of-Infection trends for Human Taenia solium Taeniasis/Cysticercosis”.

Reviewer #2 (Recommendations for the authors):Although the work is well written and very organized, I have some concerns about the model and assumptions used in the analysis.Equation 3 is different than the one reported by Diggle, 2011. One essential set of parentheses is missing. Please, clarify it.

We thank the reviewer for spotting this. We have amended equation 3 (line 555-556) as follows:p′(a) = (1 − sp) + (se + sp − 1) × p(a)

The methods section would be improved by presenting the motivations for the models' choice and a brief description of them, including assumptions and limitations. Although the reversible catalytic model (rho>0) is a generalization of the simple form (rho=0), there are two mutually exclusive assumptions: either only conversion or conversion and reversion. Because of it, unless well justified, it seems unlike that a particular pattern of an infectious disease could be described considering two fundamentally different biological processes. As observed in Figures2, 3 and 4, the data sometimes support the simple model, sometimes support the reversible, and sometimes it is not possible to decide. It means to me that it is not possible to decide between both designs which encapsulates the mechanism driving the disease age-prevalence presented by the data. Please, clarify it.

The comments raised by the reviewer are well received. There is indeed paucity of information and data available in the wider literature to understand (a) whether and where antibody seroreversion occurs; (b) the potential degree of variation in seroreversion rates between settings; (c) how long, on average, taeniasis and cysticercosis persist in humans in different epidemiological settings (estimated in our work by taking the reciprocal of infection loss rates), and (d) whether a lack of infection loss better explains trends in age-(sero)prevalence profiles. These are key knowledge gaps that are not described/explored in detail elsewhere. Our modelling approach, albeit more phenomenological rather than truly mechanistic, provides a start to identify potentially underlying drivers of epidemiological patterns. However, we agree that we should describe better our rationale / motivation to justify the approach taken to investigate two models which include these two different biological processes as follows (lines 537 – 546):

“These two model configurations represent fundamentally different processes (i.e., presence or absence of Ab-seroreversion or infection loss). Therefore, it was the intention of this analysis to explore which of these processes best captures available age-(sero)prevalence data across different epidemiological settings, given the current knowledge gaps in the literature. In particular, there is minimal knowledge relating to how long taeniasis and cysticercosis infections and seropositivity persist in human hosts, on average (estimated here by the reciprocal of infection loss and Abseroreversion rates). This enabled us to consider the possibility that transmission intensity drives the presence and extent of infection loss and Ab-seroreversion rates (due to transient antigen responses and/or partial parasite establishment, as explored in the Discussion).”

We then postulate possible explanations for a stronger degree of infection loss rates that relate to a higher proportion of transient antigen responses/partial establishment of infection in higher transmission settings in the Discussion (lines 432 – 440), and the need for further investigation into this area.

The reviewer is also correct in highlighting that across Figures 2-4 (note now Figures 3-5), there is not always a definitive answer to which biological mechanism prevails in terms of explaining age-prevalence dynamics with the datasets found for this study. This is generally a reflection of fitting these models to diverse, but often limited data sources (for example where there is minimal information particularly in the early and latter part of the curves (the very young or older groups in the population) to adequately capture the age-prevalence saturation dynamics). Flat age-(sero)prevalence profiles in low prevalence settings, not necessarily representing limited data but more a reflection of specific epidemiological conditions, also proved challenging for model fitting with these types of parsimonious models.

However, some trends emerged, particularly across human cysticercosis age(sero)prevalence profiles, in which the majority (71%) of the studies derived from the general systematic literature search indicated that the model with antibody seroreversion or infection loss (reversible model) was preferable across epidemiological settings. (This is 10 out of 14 studies, excluding the Colombian data, but it was also true for the models fitted to all Colombian departments.) This was a key observation also when models were jointly fitted across multiple datasets (e.g., LLGPEITB antibody assay and B158/B60 Ag-ELISA), therefore providing a strong signal that this mechanism (with antibody seroreversion/infection loss) may be driving the dynamics across the majority of settings.

Another concern is regarding to the saturation of the reversible model. The current formulation assumes that all individuals are exposed/susceptible to the disease and, therefore, its saturation is controlled by the reversion rate parameter (rho). The greater rho, the lower the saturation. Hypothetically, one can think of a situation in which just a fraction of the population is exposed/susceptible to an underlined disease. In this scenario, the greater the fraction of individuals not exposed/susceptible, the greater the interference bias of the reversion rate. I wonder whether the authors believe that this is a reasonable possibility and, in the affirmative case, please acknowledge the limitation of the methodology.

We thank the reviewer for raising this consideration around the behaviour of the reversible model. The catalytic model configurations as presented in this analysis assume an average FoI to which all age groups are subjected. Thus, in terms of potential exposure heterogeneity among individuals or by age, we acknowledge that the current model formulations would not be able to reflect these dynamics. Equally, other heterogeneities in exposure may exist, such as by sex (if for example household roles or occupational practices drive exposure, as the case with other NTDs). Currently there are limited or no data for *T. solium* to indicate how these heterogeneities may underpin dynamics. We discuss in some detail the limitations with the current models in capturing these more complex FoI dynamics, such as age-dependent exposure patterns, in the paragraph (lines 474 – 492) of the Discussion.